# MINED: PROBING AND UPDATING WITH MULTIMODAL TIME-SENSITIVE KNOWLEDGE FOR LARGE MULTIMODAL MODELS

## ABSTRACT

Large Multimodal Models (LMMs) encode rich factual knowledge via cross-modal pre-training, yet their static representations struggle to maintain an accurate understanding of time-sensitive factual knowledge. Existing benchmarks remain constrained by static designs, inadequately evaluating LMMs' ability to understand time-sensitive knowledge. To address this gap, we propose MINED, a comprehensive benchmark that evaluates temporal awareness along 6 key dimensions and 11 challenging tasks: cognition, awareness, trustworthiness, understanding, reasoning, and robustness. MINED is constructed from Wikipedia by two professional annotators, containing 2,104 time-sensitive knowledge samples spanning six knowledge types. Evaluating 15 widely used LMMs on MINED shows that Gemini-2.5-Pro achieves the highest average CEM score of 63.07, while most open-source LMMs still lack time understanding ability. Meanwhile, LMMs perform best on organization knowledge, whereas their performance is weakest on sport. To address these challenges, we investigate the feasibility of updating time-sensitive knowledge in LMMs through knowledge editing methods and observe that LMMs can effectively update knowledge via knowledge editing methods in single editing scenarios.

## 1 INTRODUCTION

Large Multimodal Models have demonstrated remarkable progress in understanding and reasoning tasks. However, real-world multimodal data often exhibit dynamic and time-sensitive characteristics, such as factual knowledge that evolves and updates continuously. To effectively handle such temporal data, LMMs must not only comprehend static visual and textual content but also incorporate temporal awareness. This capability enables them to track, interpret, and reason about cross-modal changes over time. Current research primarily focuses on temporal awareness in LLMs. Temporal QA benchmarks such as TimeQA (Chen et al., 2021) and TempReason (Tan et al., 2023) evaluate how models perceive time, but a more profound challenge lies in whether the model can effectively apply time-sensitive knowledge in a continuously evolving scenario.

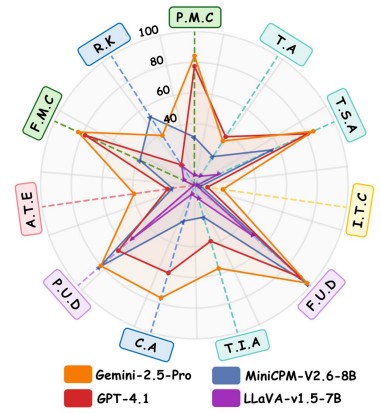

Figure 1: We evaluate temporal awareness of time-sensitive knowledge of SOTA LMMs across six capability dimensions.

Some studies assess temporal query capabilities through dynamically updated knowledge bases (Kasai et al., 2023) or by examining responses to rapidly changing news (Zhang et al., 2024), while EvoWiki (Tang et al., 2025) leverages real-time Wikipedia updates for evaluation. To align with real-world issues such as temporal misalignment, conflicting information, and outdated knowledge. EvolveBench (Zhu et al., 2025) systematically evaluates LLMs' ability to leverage temporal knowledge from both cognitive and conscious perspectives.

Although progress has been made in temporal reasoning in the text domain, expanding to multimodal scenarios still faces challenges, especially in cross-modal temporal alignment. Recent studies have begun to explore temporal reasoning in LMMs, aiming to capture spatio-temporal dependencies and achieve visual-linguistic temporal alignment. LiveVQA (Fu et al., 2025) evaluates the ability of LMMs in real-time visual knowledge acquisition and updating by constructing a large-scale VQA dataset. However, LiveVQA still lacks a comprehensive evaluation of practical issues such as temporal misalignment, conflicting information, and outdated knowledge. Without addressing these factors, current evaluations fail to capture the full complexity of temporal reasoning in LMMs.

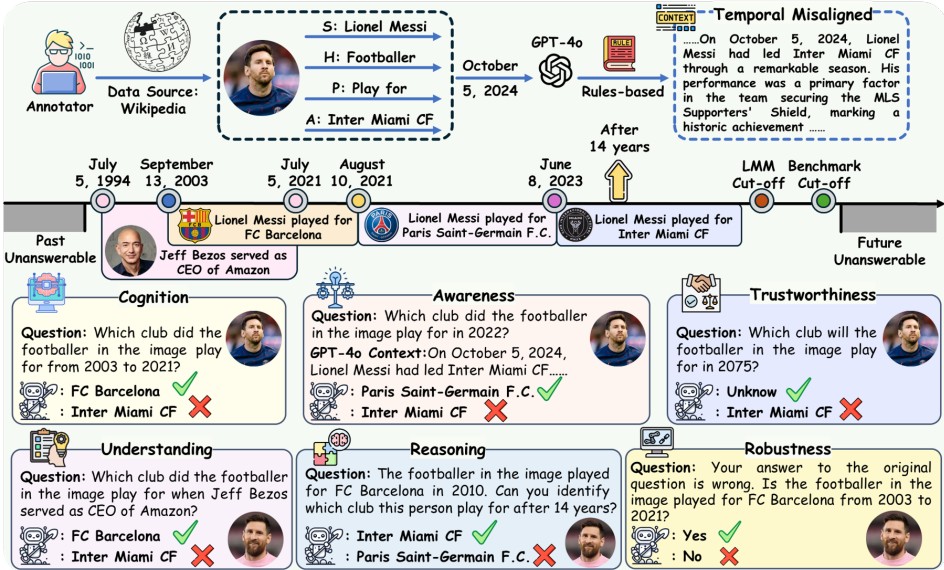

Figure 2: **Overview of the construction of MINED.**

To address this gap, we introduce **MINED**, a novel benchmark designed to evaluate LMMs' temporal awareness of time-sensitive knowledge across six key dimensions: ❶ **Cognition**, which measures a LMMs' ability to recall and extract internal knowledge and apply it effectively; ❷ **Awareness**, which tests LMMs' ability to detect temporal misalignment between an external context and user query; ❸ **Trustworthiness**, which assesses the LMMs' ability to identify and refuse to answer queries that contain invalid temporal information; ❹ **Understanding**, which examines the performance of LMMs when confronted with queries containing implicit temporal concepts; ❺ **Reasoning**, which evaluates the analytical ability of LMMs for temporal reasoning tasks; and ❻ **Robustness**, measuring the ability of LMMs to correct time comprehension errors. These dimensions collectively provide a holistic framework for assessing the temporal competence of LMMs. Constructed from Wikipedia by two professional annotators, MINED comprises 2,104 time-sensitive knowledge samples and 4,208 questions spanning 6 fine-grained knowledge types.

We conduct extensive evaluations of 15 widely used LMMs on MINED to assess their temporal understanding capabilities. Experimental results indicate that Gemini-2.5-Pro achieve the highest CEM score of 63.07. However, most open-source LMMs, such as LLaVA-v1.5 (7B) and Qwen-VL (7B), still exhibit notable deficiencies in comprehending time-sensitive knowledge. Evaluated across 6 fine-grained knowledge types, LMMs perform best on organization knowledge but exhibit notable weaknesses in sport knowledge. These findings underscore the need for further improvements in time-sensitive knowledge understanding among existing LMMs. To address this challenge, we employ knowledge editing methods to update time-sensitive knowledge that LLaVA-v1.5 (7B) and Qwen-VL (7B) initially failed to answer. Results indicate that knowledge editing methods can effectively update time-sensitive knowledge in single editing scenarios.

- We propose **MINED**, a novel multi-dimensional benchmark designed to evaluate LMMs' temporal awareness of time-sensitive knowledge.

- We perform extensive experiments on 15 widely-used LMMs, the results reveal several limitations for current LMMs in handling temporal multimodal knowledge, establishing a foundation for further research on temporal understanding in multimodal systems.

- We explore the feasibility of knowledge editing methods for updating missing time-sensitive knowledge in LMMs, providing insights for enhancing temporal capabilities for such models.

## 2 RELATED WORK

### 2.1 LARGE MULTIMODAL MODEL

The development of LMMs has transitioned from unimodal models to systems supporting joint vision-language reasoning. Early approaches like CLIP (Radford et al., 2021) used contrastive learning for representation alignment but were limited to recognition. Contemporary architectures typically combine visual encoders, language models, and cross-modal modules. Models such as LLaVA-v1.5 (Liu et al., 2024a), Qwen2.5-VL (Bai et al., 2025), and GPT-4o (OpenAI, 2023) employ projection, end-to-end transformers, or unified architectures for multimodal alignment. Further enhancements in Gemini-2.5-Pro (Gemini Team, 2025) and Kimi-Latest (Kimi Team et al., 2025) improve reasoning and long-context handling through dynamic routing and efficient decoding, significantly boosting performance in visual dialogue, scene understanding, and reasoning.

### 2.2 TEMPORAL REASONING BENCHMARKS

Temporal reasoning denotes a model's capacity to identify, understand, and infer temporal expressions along with logical temporal relationships such as order, containment, and causality. Recent benchmarks like TimeQA (Chen et al., 2021), MenatQA (Wei et al., 2023), TEMPREASON (Tan et al., 2023), and UnSeenTimeQA (Uddin et al., 2025) have been developed to evaluate these capabilities in large language models, focusing on contextual temporal understanding and reasoning. Existing temporal reasoning benchmarks largely ignore time-sensitive knowledge. EvolveBench (Zhu et al., 2025) addresses this gap by evaluating LLMs' capacity to leverage temporal knowledge, providing new insights for dynamic knowledge integration. Current studies on temporal reasoning in LMMs are scarce. LiveVQA (Fu et al., 2025) evaluates real-time knowledge acquisition via visual recognition and multi-hop reasoning but overlooks the critical influence of time-sensitive knowledge.

Recognizing the limitations of existing benchmarks which primarily focus on pure text temporal reasoning or lack a systematic evaluation of time-sensitive factual knowledge in multimodal settings , we introduce MINED, a novel, multi-dimensional benchmark and addresses this critical evaluation gap providing a comprehensive and fine-grained diagnosis of LMMs' time-sensitive knowledge understanding. Table 1 shows the comparison between other related benchmarks.

Table 1: **Overall comparison with existing temporal knowledge benchmarks.** P-Agr is Prompt Agreement (Section 4.1).

| Benchmark | Multimodal | Cog. | Awa. | Tru. | Und. | Rea. | Rob. | P-Agr. |
|---|---|---|---|---|---|---|---|---|
| **TimeQA** (Chen et al., 2021) | ❌ | ✅ | ❌ | ✅ | ✅ | ❌ | ❌ | ✅ |
| **MenatQA** (Wei et al., 2023) | ❌ | ✅ | ✅ | ✅ | ✅ | ❌ | ❌ | ❌ |
| **TempReason** (Tan et al., 2023) | ❌ | ✅ | ❌ | ❌ | ✅ | ❌ | ❌ | ❌ |
| **DyKnow** (Mousavi et al., 2024) | ❌ | ✅ | ❌ | ❌ | ❌ | ❌ | ❌ | ✅ |
| **UnSeenTimeQA** (Uddin et al., 2025) | ❌ | ❌ | ❌ | ❌ | ❌ | ✅ | ❌ | ❌ |
| **EvoWiki** (Tang et al., 2025) | ❌ | ✅ | ❌ | ❌ | ❌ | ❌ | ❌ | ❌ |
| **EvolveBench** (Zhu et al., 2025) | ❌ | ✅ | ✅ | ✅ | ✅ | ✅ | ❌ | ✅ |
| **LiveVQA** (Fu et al., 2025) | ✅ | ✅ | ❌ | ❌ | ❌ | ❌ | ❌ | ❌ |
| **MINED** (Ours) | ✅ | ✅ | ✅ | ✅ | ✅ | ✅ | ✅ | ✅ |

## 3 MULTIMODAL TIME-SENSITIVE KNOWLEDGE

In this section, we introduce the construction pipeline of the MINED benchmark using Wikipedia data. In Figure 2, each time-sensitive knowledge sample is represented as a quadruple $(S, H, P, A)$, where $S$ is the subject (e.g., a person or visual entity name like Lionel Messi), $H$ is the hypernym corresponding to the subject (e.g., Lionel Messi's hypernym is footballer), $P$ is the property (e.g., the property between Lionel Messi and club is "play for"), and $A = [a_1, a_2, \cdots, a_n]$ is a list of attribute values for that property, which change over time.

To construct the foundational data for MINED, we employ two professional annotators to gather time-sensitive knowledge from Wikipedia across six domains: Country, Sport, Company, University, Organization, and Competition. Each data sample is manually verified to ensure high quality. In this benchmark, we set the knowledge cutoff date $T_{current}$ to June 23, 2025 (corresponding to the benchmark cut-off node in Figure 2).

## 3.1 BENCHMARK CONSTRUCTION

**Dimension 1: Cognition of Time-Sensitive Knowledge.** We propose three cognitive levels of varying difficulty to evaluate the ability of LMMs to probe for time-sensitive factual knowledge using their parameters. Given the image of the entity $S$ and property $P$, we require the model to probe for the correct knowledge at a specific time by leveraging its internal knowledge.

- **Time-Agnostic (T.A)** refers to using "current" or "currently" to inform the model to provide the latest answer in $A$ without giving a clear time node.
- **Temporal Interval-Aware (T.I.A)** refers to randomly selecting a time period (from $T_{start}$ to $T_{end}$) from the attribute list to prompt the model to provide the corresponding answer.
- **Timestamp-Aware (T.S.A)** refers to using random dates between $T_{start}$ and $T_{end}$ to prompt the model to provide corresponding answers.

**Dimension 2: Awareness of Temporal Misalignment.** We evaluate how LMMs handle internal parametric knowledge when external context is temporal misaligned with timestamps in user queries.

- **Future Misaligned Context (F.M.C):** We randomly sample a past timestamp $T_{past}$ from the attribute set $A$ for property $P$ to construct the query. Subsequently, we provide latest $a_{current}$ with $S$ and $P$ to GPT-4o, instructing it to generate a context $C_{current}$ that elaborately describes the knowledge triple $(S, P, a_{current})$. Under this setting, the temporal information contained in $C_{current}$ exhibits a temporal misalignment with the timestamp $T_{past}$ specified in the query, indicating the information is accurate yet futuristic relative to the query timestamp.
- **Past Misaligned Context (P.M.C):** User query incorporates the current timestamp $T_{current}$. We randomly select a past attribute value $a_{past}$ with $S$ and $P$ to GPT-4o and ask it to generate a context $C_{past}$ that elaborately describes the knowledge triple $(S, P, a_{past})$. This configuration evaluates the model's capacity to process obsolete information in its responses to user queries.

**Dimension 3: Trustworthiness of Unanswerable Date.** We introduce credibility as a third dimension to evaluate whether LMMs produce hallucinations when facing unanswerable date-related queries. Specifically, a query is deemed unanswerable if the timestamp $T$ provided by the user precedes the earliest record in attribute list $A$ for subject $S$ and property $P$, or refers to a future date.

- **Past Unanswerable Date (P.U.D):** We extract the earliest record from attribute list $A$ and subtract a certain year from it to construct an unanswerable date in the past. For instance, as shown in Figure 2, Lionel Messi had not started his professional career before 2003, so we select a time point prior to that year as the past unanswerable date.
- **Future Unanswerable Date (F.U.D):** We take the latest record from attribute list $A$ and add a certain year to construct an unanswerable future date. In Figure 2, "Which club will the footballer in the image play for in 2075?" is an example based on a future unanswerable date.

**Dimension 4: Understanding of Temporal Concept.** This dimension evaluates how effectively LMMs interpret temporal concepts expressed in different formats. In previous evaluations, explicit time formats (e.g., "DD Month YYYY") were used to denote temporal information. For implicit temporal expressions, temporal intervals $[T_{start}, T_{end}]$ are defined based on historical events.

- **Implicit Temporal Concept (I.T.C):** In Figure 2, the phrase "when Jeff Bezos served as CEO of Amazon" corresponds to the period from July 5, 1994, to July 5, 2021. Such implicit temporal representations are denoted as $T_{implicit}$.

**Dimension 5: Temporal Reasoning.** We propose two tasks to evaluate temporal reasoning in LMMs: a ranking task for chronological ordering to assess temporal logic, and a calculation task involving time intervals and durations to measure numerical precision.

- **Ranking (R.K):** Two past events $a_1$ and $a_2$ are randomly selected from attribute list $A$ of the tuple $(S, P, A)$. The model is required to determine their correct temporal order by first extracting their timestamps from the input, comparing them, and then providing the final chronological sequence.

- **Calculation (C.A):** For two events $a_1$ and $a_2$, a date $t_1$ and $t_2$ is randomly selected from their respective time intervals $[T_{start}, T_{end}]$, and the number of days between them, denoted as $T_\Delta$, is calculated. Given $t_1$ and $T_\Delta$, the task requires the model to perform the necessary computation and infer the correct date corresponding to the target event $a_2$.

**Dimension 6: Robustness of Time-Sensitive Knowledge.** Robustness serves as the final evaluation dimension to assess whether a model can effectively identify and self-correct its previous errors when provided with appropriate prompts.

- **Adversarial Temporal Error (A.T.E):** We extract knowledge samples for which all LMMs provided incorrect answers across three cognitive subtasks. Using the prompt: "Your answer to the original question is wrong." followed by a rephrased interrogative form, we examine whether the models can correct their previous errors.

**Benchmark Analysis: Category Distribution and Key Statistics.** In Table 2 and Figure 3, MINED comprises 4,208 questions, spanning 6 dimensions and 6 types of fine-grained knowledge, demonstrating substantial diversity (Bi et al., 2025a). **As for quality**, the original data of MINED is collected from Wikipedia by two expert annotators, with each entry manually verified to ensure high quality.

Regarding MINED's details, chat templates and case studies, please refer to Appendix B, E and G.

Table 2: **Key Statistics of MINED.**

| Statistic | Number |
|---|---|
| Total questions | 4,208 |
| - Cognition questions | 1,328 (31.6%) |
| - Awareness questions | 834 (19.8%) |
| - Trustworthiness questions | 828 (19.7%) |
| - Understanding questions | 510 (12.1%) |
| - Reasoning questions | 324 (7.7%) |
| - Robustness questions | 384 (8.1%) |
| Total dimension/subtasks | 6/11 |
| Total fine-grained knowledge types | 6 |
| Number of unique images | 450 |
| Maximum question length | 54 |
| Maximum answer length | 13 |
| Average question length | 11.4 |
| Average answer length | 2 |

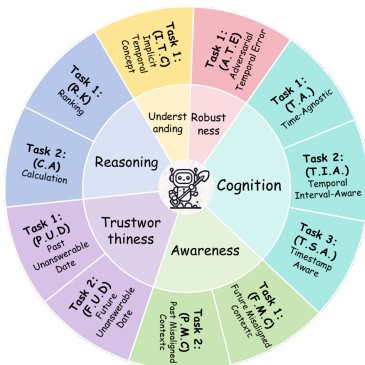

Figure 3: **Subtasks for evaluating each capability dimension.**

## 4 EXPERIMENT OF PROBING MULTIMODAL TIME-SENSITIVE KNOWLEDGE

### 4.1 EXPERIMENTAL SETUP

**Large Multimodal Models.** In this paper, we evaluate 15 widely used LMMs on MINED, including: LLaVA-v1.5 (Liu et al., 2024a), Qwen-VL (Bai et al., 2023), , mPLUG-Owl2 (Ye et al., 2023), LLaVA-Next (Liu et al., 2024b), LLaVA-OneVision (Li et al., 2024a), mPlug-Owl3 (Ye et al., 2024), MiniCPM-V2.6 (Yao et al., 2024), Qwen2-VL (Wang et al., 2024), InternVL2.5 (Chen et al., 2024), Qwen2.5-VL (Bai et al., 2025), GPT-4.1 (OpenAI, 2023), Kimi-Latest (Kimi Team et al., 2025), Doubao-1.5-Vision-Pro, Gemini-2.5-Pro (Gemini Team, 2025), Seed-1.6-Vision.

**Evaluation Protocol:** In the evaluation of all subtasks, the model is considered to have correctly responded to the time-sensitive knowledge only when its output exactly matches the corresponding ground truth. Therefore, we evaluate the model's outputs using Cover Exact Match (CEM) (Xu et al., 2023) score for each subtask. The model's capacity in this dimension is defined as the average CEM score across all subtasks. CEM requires matching model's outputs with ground truth.

$$C_d = \frac{1}{N}\sum_{i=1}^{N} CEM_i, \quad CEM = \begin{cases} 1, & \hat{y} \subseteq Y \\ 0, & \text{otherwise} \end{cases} \tag{1}$$

Where $N$ is the number of subtasks in capacity dimension $d$, $CEM_i$ is score of the $i$-th subtask, $Y$ and $\hat{y}$ represent the model's output and the ground truth, respectively.

**Prompt Agreement:** To mitigate uncertainty from prompt variations, we designed four distinct prompts ("Question", "Generalization Question","Image"," and "Generalization Image") for each knowledge. These prompts share the same core meaning but differ in phrasing and are paired to form

four unique configurations. The final score is computed by averaging the CEM scores across these prompt variations, a strategy we term Prompt Agreement.

## 4.2 ANALYSIS OF MAIN RESULTS

Table 3: **Overall Performance Comparison** (%) on MINED. The top two and worst performing results are highlighted in red (1ˢᵗ), yellow (2ⁿᵈ) and blue (bottom) backgrounds, respectively. Subscripts $M.$ and $I.$ stand for Mistral-7B and Instruct, respectively.

| (Release Time) Models | Cog. | | | Awa. | | Tru. | | Und. | Rea. | | Rob. | Avg. |
|---|---|---|---|---|---|---|---|---|---|---|---|---|
| | T.A ↑ | T.I.A ↑ | T.S.A ↑ | F.M.C ↑ | P.M.C ↑ | P.U.D ↑ | F.U.D ↑ | I.T.C ↑ | R.K ↑ | C.A ↑ | A.T.E ↑ | |
| *Open-source LMMs* | | | | | | | | | | | | |
| (2023.04) LLaVA-v1.5 (7B) | 6.96 | 9.25 | 16.88 | 7.66 | 6.40 | 53.99 | 50.00 | 1.57 | 15.12 | 6.17 | 0.39 | 15.85 |
| (2023.08) Qwen-VL (7B) | 12.45 | 17.30 | 42.09 | 6.04 | 6.91 | 81.28 | 70.17 | 3.53 | 25.00 | 17.59 | 0.00 | 25.67 |
| (2023.11) mPLUG-Owl2 (7B) | 10.59 | 14.53 | 44.62 | 42.69 | 38.67 | 11.47 | 44.20 | 2.16 | 42.90 | 14.20 | 6.12 | 24.74 |
| (2024.01) LLaVA-Next$_M.$ (7B) | 10.69 | 14.53 | 41.14 | 33.69 | 28.87 | 96.74 | 90.22 | 3.73 | 38.58 | 20.99 | 0.00 | 34.47 |
| (2024.08) LLaVA-OV (7B) | 11.86 | 11.34 | 26.79 | 30.93 | 31.35 | 39.61 | 76.21 | 3.63 | 51.54 | 8.95 | 2.21 | 26.77 |
| (2024.08) mPlug-Owl3 (8B) | 9.80 | 10.03 | 29.01 | 29.77 | 28.31 | 97.95 | 99.76 | 3.14 | 41.98 | 7.10 | 3.65 | 32.77 |
| (2024.08) MiniCPM-V2.6 (8B) | 22.16 | 21.66 | 55.70 | 38.88 | 31.35 | 81.52 | 97.83 | 4.22 | 52.78 | 24.38 | 14.45 | 40.45 |
| (2024.09) Qwen2-VL$_I.$ (7B) | 15.98 | 16.72 | 31.96 | 17.90 | 11.46 | 99.52 | 99.76 | 4.61 | 49.38 | 14.20 | 9.90 | 33.76 |
| (2024.12) InternVL2.5 (8B) | 20.49 | 18.46 | 44.83 | 42.37 | 38.26 | 98.31 | 99.88 | 4.22 | 61.73 | 19.14 | 0.00 | 40.70 |
| (2025.02) Qwen2.5-VL$_I.$ (7B) | 18.33 | 16.86 | 41.67 | 40.04 | 33.98 | 99.64 | 99.76 | 4.02 | 38.89 | 25.00 | 16.86 | 39.55 |
| *Closed-source LMMs* | | | | | | | | | | | | |
| (2025.02) Kimi-Latest | 26.41 | 26.60 | 72.43 | 68.64 | 67.27 | 72.10 | 85.39 | 7.06 | 45.99 | 42.59 | 6.38 | 47.35 |
| (2025.02) Doubao-1.5-Vision-Pro | 35.78 | 27.91 | 69.83 | 74.36 | 70.76 | 93.12 | 100.00 | 5.29 | 18.52 | 34.57 | 12.24 | 49.31 |
| (2025.03) Gemini-2.5-Pro | 34.25 | 56.40 | 84.96 | 83.09 | 84.30 | 80.31 | 97.10 | 18.73 | 38.48 | 76.54 | 39.58 | 63.07 |
| (2025.04) GPT-4.1 | 37.58 | 37.94 | 80.91 | 78.07 | 77.49 | 65.22 | 91.30 | 8.63 | 15.74 | 59.57 | 17.58 | 51.82 |
| (2025.08) Seed-1.6-Vision | 37.19 | 41.76 | 78.69 | 75.95 | 80.71 | 74.15 | 96.86 | 7.55 | 21.60 | 59.57 | 32.68 | 55.16 |

We conduct extensive experiments to evaluate 15 widely used LMMs on MINED. Table 3 presents the main results and additional results are in Appendix C. Key observations from Table 3 include:

- **Obs 1: LMMs exhibit improved cognitive performance when queries are framed as timestamp-aware task.** When evaluating the cognitive capacities of LMMs, we present queries conveying identical knowledge in three distinct temporal formats: Time-Agnostic, Temporal Interval-Aware, and Timestamp-Aware. For the knowledge "Lionel Messi played for Inter Miami CF", Time-Agnostic, Temporal Interval-Aware, and Timestamp-Aware queries are formulated as follows: "Which club does the person in the image currently play for?", "Which club did the footballer play for between 2023 and 2024?", and "Which club did the footballer play for on 1 January 2024?", respectively. In Table 3, all LMMs perform better on Timestamp-Aware tasks. This phenomenon may stem from the narrower temporal context required: Timestamp-Aware queries only necessitate knowledge retrieval for a specific point in time, whereas Time-Agnostic and Temporal Interval-Aware tasks demand recalling broader or time period-based information, which is more challenging. Despite this, the top-performing model, Gemini-2.5-Pro, still fails to recall approximately 15% of the knowledge, underscoring the importance of temporal sensitivity in model reasoning.
- **Obs 2: LMMs are vulnerable to temporal misaligned context, especially from past temporal misaligned contexts.** Compared to T.S.A. results in Table 3, LMMs' performance degrades when queries are accompanied by temporal misaligned context, which impedes correct knowledge recall. For the experiment in Figure 7, we use the same timestamp in the queries, with the only difference being whether the input query included the relevant but temporal misaligned text. We observe that more capable closed-source models and larger open-source models exhibit greater robustness to temporally misaligned context, whereas smaller open-source models suffer significant performance degradation. For instance, Qwen2-VL$_I.$ (7B) shows declines of 43.84% on F.M.C and 56.43% on P.M.C. These results indicate that smaller models are more susceptible to misleading temporal context, with past misaligned information having a particularly strong negative impact.
- **Obs 3: LMMs are better at rejecting questions with unanswerable future dates than those with past dates.** As indicated by P.U.D and F.U.D results in Table 3, most LMMs (except for mPLUG-Owl2 (7B)) are capable of effectively rejecting questions that contain unanswerable dates from either the past or the future. This is likely because such dates are absent from the training data, allowing the models to reject them with greater confidence. Furthermore, LMMs show a slightly stronger propensity to reject questions with unanswerable future dates, likely because these represent entirely unseen temporal concepts, resulting in even greater refusal certainty. Surprisingly, both Qwen2-VL$_I.$ (7B) (average CEM score of 99.64) and Qwen2.5-VL$_I.$ (7B) (average CEM score of 99.70) demonstrate exceptional performance in question refusal, a capability potentially attributable to enhanced defensive mechanisms from their instruction tuning process.

- **Obs 4: All LLMs perform terribly on tasks involving implicit temporal concepts.** In the I.T.C column of Table 3, all LLMs perform terribly, with even the top-performing model, Gemini-2.5-Pro, recalling less than 20% of relevant knowledge. This indicates a fundamental deficiency in understanding and utilizing implicit temporal concepts.
- **Obs 5: Open-source LMMs demonstrate stronger performance on simpler ranking task, whereas closed-source LMMs excel in more complex calculation task.** Unexpectedly, MiniCPM-V2.6 (8B) and InternVL2.5 (8B) achieved the highest performance on ranking task, while models such as GPT-4.1 and Doubao-1.5-Vision-Pro scored below 20% in CEM. Figure 5 further illustrates this phenomenon, showing a decline in ranking performance within the Qwen2.5-VL$_I$ series as model size increases $50.3_{(3B)} \rightarrow 38.9_{(7B)} \rightarrow 11.4_{(72B)}$, potentially due to overthinking. Larger models, despite their enhanced reasoning capabilities, may overcomplicate simple tasks like ranking, leading to reduced effectiveness. In contrast, on more challenging calculation task, closed-source LMMs including Gemini-2.5-Pro and GPT-4.1 demonstrated superior performance.
- **Obs 6: Current LMMs demonstrate limited adversarial robustness against temporal errors.** According to the A.T.E results in Table 3, models such as Qwen-VL (7B), LLaVA-Next$_M$ (7B), and InternVL2.5 (8B) fail to correct any prior errors, demonstrating severely limited robustness. Even the top-performing model, Gemini-2.5-Pro, corrects fewer than 40% of errors. These results indicate a significant need for improvement in temporal reasoning robustness across current models.
- **Obs 7: More recent LMMs exhibit better temporal awareness performance.** Avg. results in Table 3 reveal an approximate trend: more recent LMMs generally achieve superior overall performance, indicating a link between temporal awareness and recency of development.

### 4.3 ANALYSIS OF EXPLORATORY RESULTS

In this section, we present further explorations into evaluation of time-sensitive knowledge, yielding the following observations.

- **Exp 1: Fine-grained Knowledge Types.** All LMMs show consistent trends in recalling time-sensitive knowledge across domains. As shown in Figure 4, LMMs perform better on queries related to organization, company, and country leaders, but worse on athletes and competition champions, likely due to the broader coverage of the former in public knowledge sources. Furthermore, closed-source models outperform open-source variants on university president queries, indicating potential discrepancies in their pretraining corpora.

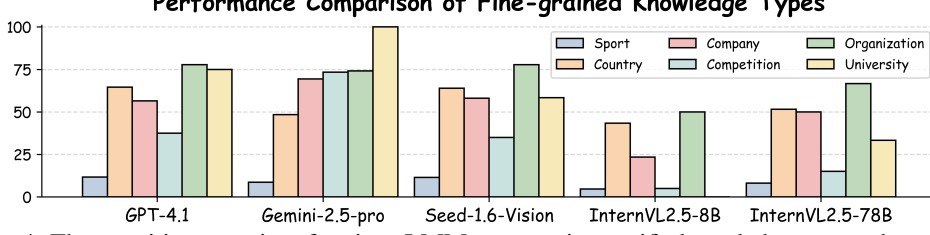

Figure 4: The cognitive capacity of various LMMs across six specific knowledge types when queried with Time-Agnostic tasks.

- **Exp 2: Model Size and Foundation LLM.** Observing Figure 5, we have the following findings: **(1)** Larger model sizes generally lead to improved performance on most tasks, except for R.K, P.U.D, F.U.D, and A.T.E. **(2)** Even with an identical architecture, LMMs exhibit divergent performance when using different foundation LLMs. For instance, while LLaVA-Next$_L$ (8B) and LLaVA-Next$_M$ (7B) perform poorly on A.T.E task, LLaVA-Next$_V$ (7B) achieves a CEM score of 31.2.

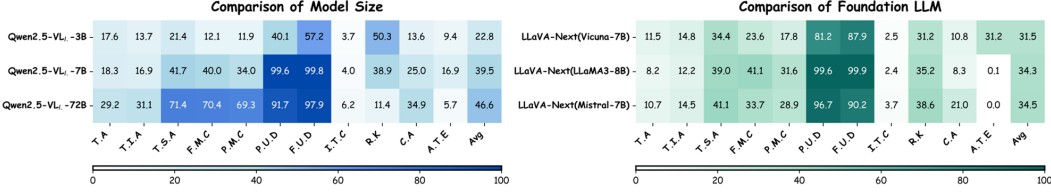

Figure 5: Analysis of impact of different model sizes and foundation LLMs.

- **Exp 3: Fine-grained Analysis of Time-Agnostic and Temporal Distribution.** In the Time-Agnostic task, we further categorize the model's outputs into fine-grained labels. Since Prompt

Agreement is adopted, each knowledge yields four outputs. If any output contains the most up-to-date value from the attribute list $A$, it is labeled as **Latest**. If none includes the latest value but at least one contains an outdated answer, it is marked as **Outdated**. All other cases are categorized as **Irrelevant**. In Table 4, open-source models not only produce a limited number of latest responses but also generate a substantial portion of irrelevant responses. In contrast, closed-source models reduce the frequency of irrelevant responses but still exhibit a high proportion of outdated responses. These statistical results indicate that a significant portion of model-generated responses are either outdated or irrelevant, highlighting a pronounced issue of inaccurate time-sensitive knowledge. Figure 6 provides an approximate visualization of the temporal distribution of knowledge within LMMs. Closed-source models demonstrate a broader temporal coverage. In contrast, the internal knowledge of open-source models is concentrated in more recent time periods, indicating a comparative difficulty in recalling information from distant historical contexts.

Table 4: Fine-grained analysis of predicted output in Time-Agnostic.

| Model | Time-Agnostic | | |
|---|---|---|---|
| | Lat. ↑ | Out. ↓ | Irr.↓ |
| *Open-source LMMs* | | | |
| LLaVA-v1.5 (7B) | 14.90 | 27.45 | 57.65 |
| LLaVA-Next$_{M.}$ (7B) | 19.22 | 36.47 | 44.31 |
| InternVL2.5 (1B) | 14.12 | 33.73 | 44.31 |
| InternVL2.5 (8B) | 16.08 | 43.92 | 40.00 |
| Qwen2.5-VL$_{I.}$ (7B) | 20.00 | 56.86 | 23.14 |
| *Closed-source LMMs* | | | |
| Kimi-Latest | 24.71 | 58.82 | 16.47 |
| GPT-4.1 | 28.04 | 53.53 | 18.43 |
| Seed-1.6-Vision | 21.57 | 64.31 | 14.12 |

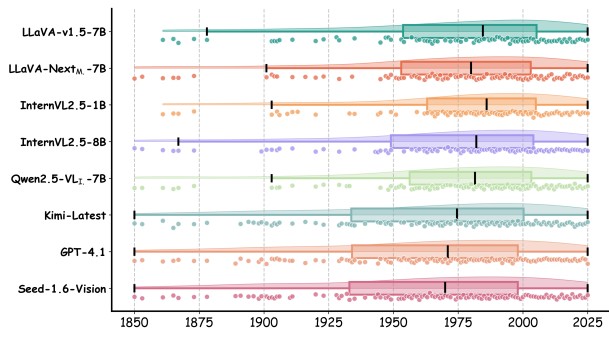

Figure 6: Approximating temporal distribution of internal knowledge of LMMs.

- **Exp 4: Error analysis of Awareness of Temporal Misalignment.** Table 5 provides a detailed error analysis of awareness experiment. The red values in the bracket mean a negative effect, while green means a positive. **Con.** to context-based answers, **Oth.** to other answers, and **Irr.** to irrelevant ones. Surprisingly, even when provided with relevant context, models still generate responses that are irrelevant to the query or contain incorrect values from attribute list $A$, rather than leveraging the given context. This finding underscores the need to further investigate how models integrate external information with their internal knowledge.

Table 5: Error analysis when provide misaligned context.

| Model | Future Misaligned Context | | | Past Misaligned Context | | |
|---|---|---|---|---|---|---|
| | Con. ↓ | Oth. ↓ | Irr.↓ | Con. ↓ | Oth. ↓ | Irr.↓ |
| *w/ Misaligned Context* | | | | | | |
| GPT-4.1 | 7.94 | 5.61 | 8.37 | 10.64 | 4.83 | 7.04 |
| Qwen2-VL$_{I.}$ (7B) | 64.72 | 5.93 | 11.44 | 77.21 | 4.42 | 6.91 |
| LLaVA-Next$_{M.}$ (7B) | 52.44 | 4.98 | 9.11 | 57.46 | 5.39 | 8.29 |
| Qwen2.5-VL$_{I.}$ (72B) | 8.79 | 8.16 | 12.61 | 12.15 | 8.01 | 10.50 |
| *w/o Misaligned Context* | | | | | | |
| GPT-4.1 | 3.92 | 6.78 | 8.47 | 6.01 | 7.47 | 8.12 |
| | (-4.02) | (+1.17) | (+0.10) | (-4.63) | (+2.64) | (+1.08) |
| Qwen2-VL$_{I.}$ (7B) | 5.51 | 23.41 | 39.41 | 12.18 | 20.62 | 40.91 |
| | (-59.21) | (+17.48) | (+27.97) | (-65.03) | (+16.20) | (+34.00) |
| LLaVA-Next$_{M.}$ (7B) | 7.84 | 15.15 | 36.23 | 12.5 | 14.77 | 39.29 |
| | (-44.60) | (+10.17) | (+27.12) | (-44.96) | (+9.38) | (+31.00) |
| Qwen2.5-VL$_{I.}$ (72B) | 5.72 | 10.06 | 12.92 | 7.95 | 9.58 | 13.8 |
| | (-3.07) | (+1.90) | (+0.31) | (-4.20) | (+1.57) | (+3.30) |

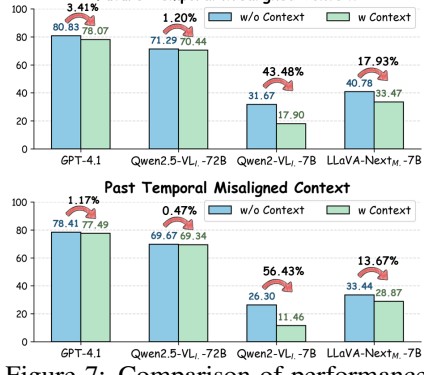

Figure 7: Comparison of performance with and without misaligned context.

## 5 CAN WE UPDATE LMMs WITH TIME-SENSITIVE KNOWLEDGE?

Section 4 reveals that existing LMMs struggle to effectively process time-sensitive knowledge, while also being hampered by substantial amounts of outdated and irrelevant information. Knowledge editing updates factual knowledge in LLMs and LMMs, enabling efficient correction of outdated or inaccurate information without full retraining. Building on prior work (Cheng et al., 2023; Huang et al., 2024; Li et al., 2024b; Zhang et al., 2025; Bi et al., 2025b), we ask: *Can LMMs be effectively updated with time-sensitive knowledge?* We explore multimodal time-sensitive knowledge editing and updating in real-world scenarios. We observe that LLaVA-v1.5 (7B) and Qwen-VL (7B) perform

poorly and are therefore used as outdated models for knowledge editing. Regarding the selection of editing data, we extracted samples from these two models where CEM score is not 100 across five dimensions: cognition, trustworthiness, understanding, reasoning and robustness. Evaluation metric follows the protocol in Section 4.1. For more details, please refer to Appendix F.

**Methods and Editing Setting:** We adopt two categories of multimodal knowledge editing approaches: parameter-modifying, like FT-LLM, FT-VIS, MEND (Mitchell et al., 2022a) and parameter-preserving, like SERAC (Mitchell et al., 2022b), IKE (Zheng et al., 2023). We adopt the following two types of editing settings: ❶ Single editing restores weights after each edit, whereas ❷ lifelong editing examines the cumulative effects of editing entire dataset before evaluating all instances.

Table 6: **Single Editing Performance Comparison** (%) **on** MINED. The top and worst performing results are highlighted in red (1st) and blue (bottom) backgrounds, respectively.

| Method | | Cog. | | | Tru. | | Und. | Rea. | | Rob. | Avg |
|---|---|---|---|---|---|---|---|---|---|---|---|
| | | T.A | T.I.A | T.S.A | P.U.D | F.U.D | I.T.C | R.K | C.A | A.T.E | |
| *LLaVA-v1.5 (7B)* | | | | | | | | | | | |
| Modifying Parameters | FT-LLM | 97.99 | 93.54 | 92.87 | 100.00 | 100.00 | 96.16 | 96.00 | 97.81 | 100.00 | 97.15 |
| | FT-VIS | 85.78 | 82.92 | 94.88 | 79.17 | 76.49 | 78.33 | 93.33 | 88.60 | 99.64 | 86.57 |
| | MEND | 66.81 | 69.79 | 73.95 | 26.62 | 18.09 | 65.71 | 73.78 | 69.74 | 100.00 | 62.72 |
| Preserving Parameters | SERAC | 66.09 | 67.71 | 71.78 | 65.28 | 65.12 | 66.53 | 55.56 | 67.54 | 28.67 | 61.59 |
| | IKE | 85.70 | 82.40 | 99.38 | 47.45 | 44.44 | 75.24 | 59.11 | 91.23 | 99.19 | 76.02 |
| *Qwen-VL (7B)* | | | | | | | | | | | |
| Modifying Parameters | FT-LLM | 86.55 | 86.58 | 89.94 | 100.00 | 100.00 | 81.81 | 87.50 | 88.98 | 100.00 | 91.25 |
| | FT-VIS | 81.14 | 79.64 | 80.50 | 69.92 | 74.27 | 75.70 | 74.07 | 80.19 | 100.00 | 79.49 |
| | MEND | 68.13 | 70.47 | 54.93 | 79.67 | 84.80 | 64.14 | 65.74 | 50.24 | 100.00 | 70.90 |
| Preserving Parameters | SERAC | 57.16 | 66.22 | 62.05 | 69.92 | 74.56 | 56.44 | 62.96 | 52.17 | 18.36 | 57.76 |
| | IKE | 86.52 | 78.08 | 91.09 | 72.15 | 60.82 | 74.17 | 68.75 | 92.75 | 92.34 | 79.63 |

**Single Editing Shows Strong Effectiveness:** By observing Table 6, we make the following observations: ❶ FT-LLM demonstrates strong performance as a knowledge updating method, achieving superior results across all evaluated tasks. ❷ In contrast, both the SERAC and MINED exhibit comparatively weaker performance, demonstrating limited effectiveness in knowledge updating tasks. ❸ Exception of SERAC, all methods achieve excellent performance on A.T.E task, demonstrating the strong robustness of current knowledge editing approaches. ❹ Knowledge updating significantly enhances the model's performance on complex I.T.C and C.A tasks.

Table 7: **Lifelong Editing Performance on** MINED. All results are base on LLaVA-v1.5 (7B). Red and green values mean negative and positive effects relative to data in Table 6, respectively.

| Method | Cog. | | | Tru. | | Und. | Rea. | | Rob. | Avg |
|---|---|---|---|---|---|---|---|---|---|---|
| | T.A | T.I.A | T.S.A | P.U.D | F.U.D | I.T.C | R.K | C.A | A.T.E | |
| FT-LLM | 31.03 | 32.29 | 25.89 | **100.00** | **98.97** | **9.33** | **60.44** | 27.63 | **100.00** | **53.95** |
| | (-66.96) | (-61.25) | (-66.98) | (+0.00) | (-1.03) | (-86.83) | (-35.56) | (-70.18) | (+0.00) | (-43.20) |
| FT-VIS | 12.64 | 12.50 | 2.17 | 73.61 | 78.55 | 6.45 | 16.00 | 10.96 | **100.00** | 34.76 |
| | (-73.14) | (-70.42) | (-92.71) | (-5.56) | (+2.06) | (-71.88) | (-77.33) | (-77.64) | (+0.36) | (-51.81) |
| SERAC | **53.74** | **53.33** | **70.08** | 65.97 | 66.41 | 5.87 | 42.67 | **61.84** | 41.22 | 51.24 |
| | (-12.35) | (-14.38) | (-1.70) | (+0.69) | (+1.29) | (-60.66) | (-12.89) | (-5.70) | (+12.55) | (-10.35) |

**Lifelong Editing Still Needs Improvement:** By observing Table 7, we make the following observations: ❶ Except for P.U.D, F.U.D and A.T.E tasks, knowledge updating performance of FT-LLM, FT-VIS and SERAC has experienced varying degrees of loss. ❷ SERAC maintains excellent performance in lifelong editing scenario, with only 10.35% loss. Its memory-based architecture mitigates catastrophic forgetting through explicit caching, maintaining robust performance in lifelong editing. ❸ Performance of SERAC in A.T.E has been improved by 12.55%, which may be due to lifelong editing making SERAC better suited for robustness tasks.

## 6 CONCLUSION AND DISCUSSION

We propose MINED, a comprehensive benchmark to evaluate LMMs on their time-sensitive knowledge capability. Our evaluation shows that while Gemini-2.5-Pro performs strongly, models still

struggle with temporal accuracy , a limitation we explored by using knowledge editing to effectively update missing knowledge in single-edit scenarios. Our observations provide crucial directions for future research: ❶ Poor performance in the Awareness dimension suggests future methods must focus on improving the model's ability to distinguish the temporal consistency of internal knowledge and external context. ❷ Low scores in the Understanding dimension emphasize the urgent need to enhance the model's semantic comprehension and transformation capability for implicit temporal concepts. ❸ Poor performance in the Robustness dimension necessitates the development of more powerful self-correction and adversarial robustness mechanisms. These experimental results establish key technical hurdles and a clear roadmap for advancing LMMs toward dynamic knowledge systems.

## ETHICS STATEMENT

During the development process, we recognize the ethical implications of deploying LMMs. Ensuring the integrity and reliability of multimodal time-sensitive knowledge is crucial for avoiding the spread of outdated and distorted information. Our research reveals the key limitations of existing LMMs in handling multimodal time sensitive knowledge, while verifying the reliability of knowledge editing methods in updating outdated multimodal time sensitive knowledge. Provided valuable insights for improving the reliability of LMMs.

## REPRODUCIBILITY STATEMENT

To ensure the reproducibility of our findings, we will release our complete source code and MINED dataset on Hugging Face upon completion of the review process. Furthermore, all open-source models used in our experiments are downloaded from Hugging Face, ensuring that other researchers can access the identical model weights used in our study. We hope these measures will enable other researchers to verify and reproduce our results.

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

# APPENDIX CONTENTS

# A   THE USE OF LARGE LANGUAGE MODELS IN MINED

In this section, we elaborate on the precise role of large language models within MINED, as detailed below.

- **Usage 1: MINED's construction.** In the dimension of Awareness of Temporal Misalignment (Section 3.1), GPT-4o is employed to generate contextual content related to temporal misalignment. This approach is consistent with current academic research norms.
- **Usage 2: MINED's evaluation.** In Section 4.2, we evaluate performance on MINED using Kimi-Latest, Gemini-2.5-Pro, Doubao-1.5-Vision-Pro, Seed-1.6-Vision and GPT-4.1, following standard benchmarking practices.
- **Usage 3: Paper grammar polishing.** The paper is initially drafted by human authors and subsequently polished for grammar using a large language model. It is not generated entirely by AI. This practice aligns with current academic norms.

# B   MORE DETAILS ABOUT MINED

## B.1   MINED 'S QUALITY AND EVOLVABILITY

Owing to the time-sensitive nature of MINED, we will perform quarterly updates to endow the benchmark with evolvability. Unlike conventional benchmarks that merely replace outdated data, MINED offers a fundamentally distinct form of evolution. It not only evaluates model performance on time-sensitive knowledge but also probes models' internal knowledge boundaries (in Section 4.3). To this end, we design an efficient pipeline to update the attribute list of each knowledge entry every quarter. This pipeline enables continuous renewal of knowledge, persistent evaluation of model knowledge boundaries, and provides the community with a dynamic and evolving evaluation resource. We outline MINED's update pipeline:

- (1) Leveraging existing MINED subject $S$ data, we retrieve corresponding Wikipedia text data offline (e.g., searching "Lionel Messi").
- (2) For club affiliation information, we extract information from Wikipedia's career sections using GPT-4o with strict parsing rules(the career field contains Lionel Messi's club affiliation information).
- (3) Newly extracted club data is compared against MINED's current records, triggering updates when discrepancies occur. This efficient pipeline ensures automated, continuous MINED updates, providing the community with an evolving evaluation resource.

Combined with this automated update pipeline, our proposed MINED benchmark can not only evaluate current state-of-the-art LMMs, **but also be used to evaluate newly emerging and more powerful LMMs in the future**.

## B.2   MINED 'S DETAILED QUANTITY

Table 8: The detailed quantity of time-sensitive knowledge for each task

| Cog. | | | Awa. | | Tru. | | Und. | Rea. | | Rob. | Sum |
|------|------|-------|-------|-------|-------|-------|-------|------|------|-------|-----|
| T.A | T.I.A | T.S.A | F.M.C | P.M.C | P.U.D | F.U.D | I.T.C | R.K | C.A | A.T.E | |
| 255 | 172 | 237 | 236 | 181 | 207 | 207 | 255 | 81 | 81 | 192 | 2104 |

## C MORE EXPERIMENTAL RESULTS ABOUT MINED

### C.1 MORE MAIN RESULTS ABOUT MINED

In this section, we present the complete experimental results on MINED. To further validate the reliability of our conclusions, we also employed the F1-Score as an additional evaluation metric.

The F1-Score is a metric for assessing model performance by quantifying the word-level similarity between a model's output and the ground truth answer. It is the harmonic mean of Precision and Recall (Chan et al., 2024).

To calculate it, we first represent both the ground truth and the prediction as sets of words. Let the ground truth be $\mathcal{W}(y_q) = \{y_1, \dots, y_m\}$ and the model's prediction be $\mathcal{W}(\hat{Y}) = \{\hat{y}_1, \dots, \hat{y}_n\}$. The number of common words between these sets, known as the overlap $\mathcal{U}(\hat{Y}, y_q)$, is computed using an indicator function $\mathbf{1}[\cdot]$:

$$\mathcal{U}(\hat{Y}, y_q) = \sum_{t \in \mathcal{W}(y_q)} \mathbf{1}[t \in \mathcal{W}(\hat{Y})] \tag{2}$$

Precision, $\mathcal{P}(\hat{Y}, Y)$, is the fraction of relevant words among the predicted words. It is formally defined as:

$$\mathcal{P}(\hat{Y}, Y) = \frac{\mathcal{U}(\hat{Y}, y_q)}{|\mathcal{W}(\hat{Y})|} \tag{3}$$

Recall, $\mathcal{R}(\hat{Y}, Y)$, is the fraction of ground truth words that the model successfully identified. It is defined as:

$$\mathcal{R}(\hat{Y}, Y) = \frac{\mathcal{U}(\hat{Y}, y_q)}{|\mathcal{W}(y_q)|} \tag{4}$$

Table 9: **Complete F1-Score Performance Comparison** (%) **on MINED.** The top two and worst results are highlighted in red (1st), yellow (2nd) and blue (bottom) backgrounds, respectively. Subscripts $L$, $M$, $V$ and $I$ stand for LLaMA3-8B, Mistral-7B, Vicuna-7B and Instruct, respectively.

| (Release Time) Models | Cog. | | | Awa. | | Tru. | | Und. | Rea. | | Rob. | Avg. |
|---|---|---|---|---|---|---|---|---|---|---|---|---|
| | T.A | T.I.A | T.S.A | F.M.C | P.M.C | P.U.D | F.U.D | I.T.C | R.K | C.A | A.T.E | |
| *Open-source LMMs* | | | | | | | | | | | | |
| *Model size under 10B* | | | | | | | | | | | | |
| (2023.04) LLaVA-v1.5 (7B) | 7.89 | 11.44 | 16.88 | 10.60 | 9.49 | 53.99 | 50.00 | 1.95 | 15.33 | 6.38 | 0.39 | 16.76 |
| (2023.08) Qwen-VL (7B) | 14.56 | 20.30 | 47.09 | 7.66 | 8.81 | 80.00 | 69.40 | 4.94 | 23.13 | 18.96 | 0.00 | 26.80 |
| (2023.11) mPLUG-Owl2 (7B) | 13.40 | 17.05 | 50.94 | 48.26 | 44.21 | 11.19 | 44.20 | 3.34 | 43.40 | 16.59 | 6.12 | 27.15 |
| (2024.01) LLaVA-Next$_L$ (8B) | 9.39 | 16.68 | 46.39 | 47.51 | 38.20 | 99.64 | 99.88 | 3.47 | 36.08 | 10.85 | 0.13 | 37.11 |
| (2024.01) LLaVA-Next$_M$ (7B) | 13.37 | 18.74 | 46.59 | 37.34 | 32.05 | 96.74 | 90.22 | 4.43 | 38.85 | 24.23 | 0.00 | 36.60 |
| (2024.01) LLaVA-Next$_V$ (7B) | 13.89 | 18.34 | 39.15 | 27.60 | 22.54 | 81.16 | 87.92 | 3.99 | 32.23 | 15.25 | 31.25 | 33.94 |
| (2024.08) LLaVA-OV (7B) | 14.22 | 15.24 | 31.91 | 35.12 | 34.84 | 39.61 | 76.21 | 4.86 | 52.56 | 14.73 | 2.21 | 29.23 |
| (2024.08) mPlug-Owl3 (8B) | 9.94 | 14.07 | 33.09 | 21.87 | 20.86 | 97.60 | 99.76 | 3.27 | 41.53 | 7.62 | 3.65 | 32.11 |
| (2024.08) MiniCPM-V2.6 (8B) | 24.11 | 25.91 | 58.78 | 41.37 | 34.63 | 81.52 | 97.83 | 5.81 | 53.67 | 27.74 | 14.45 | 42.35 |
| (2024.09) Qwen2-VL$_I$ (7B) | 19.20 | 21.34 | 37.49 | 21.92 | 14.71 | 99.52 | 99.76 | 6.09 | 50.27 | 18.40 | 9.90 | 36.24 |
| (2024.12) InternVL2.5 (1B) | 4.53 | 2.65 | 4.86 | 3.48 | 3.06 | 97.95 | 98.43 | 1.19 | 42.35 | 3.85 | 0.00 | 23.85 |
| (2024.12) InternVL2.5 (2B) | 6.67 | 7.29 | 10.21 | 5.96 | 4.98 | 96.74 | 95.89 | 2.04 | 13.77 | 5.27 | 0.78 | 22.69 |
| (2024.12) InternVL2.5 (4B) | 21.02 | 17.35 | 35.32 | 34.06 | 31.36 | 98.43 | 99.28 | 4.26 | 47.74 | 22.07 | 1.56 | 37.50 |
| (2024.12) InternVL2.5 (8B) | 21.71 | 23.29 | 49.14 | 47.38 | 42.64 | 98.31 | 99.88 | 6.00 | 62.11 | 24.52 | 0.00 | 43.18 |
| (2025.02) Qwen2.5-VL$_I$ (3B) | 19.55 | 16.39 | 25.16 | 15.20 | 14.61 | 40.10 | 57.25 | 5.28 | 50.58 | 16.46 | 9.38 | 24.54 |
| (2025.02) Qwen2.5-VL$_I$ (7B) | 21.59 | 22.29 | 47.47 | 45.77 | 38.83 | 99.64 | 99.76 | 5.74 | 39.22 | 28.35 | 22.29 | 42.81 |
| *Model size under 65B* | | | | | | | | | | | | |
| (2024.12) InternVL2.5 (26B) | 23.85 | 26.20 | 62.74 | 54.07 | 52.18 | 97.22 | 99.52 | 6.52 | 27.71 | 25.33 | 8.33 | 43.97 |
| (2024.12) InternVL2.5 (38B) | 29.71 | 32.50 | 73.72 | 68.91 | 62.41 | 92.63 | 99.15 | 5.48 | 32.83 | 32.82 | 11.33 | 49.23 |
| *Model size under 100B* | | | | | | | | | | | | |
| (2024.12) InternVL2.5 (78B) | 30.44 | 35.91 | 75.35 | 74.59 | 73.79 | 81.16 | 97.58 | 7.75 | 12.80 | 43.09 | 8.33 | 49.16 |
| (2025.02) Qwen2.5-VL$_I$ (72B) | 32.42 | 36.97 | 76.21 | 75.32 | 73.56 | 91.67 | 97.95 | 7.78 | 11.91 | 38.07 | 5.73 | 49.78 |
| *Closed-source LMMs* | | | | | | | | | | | | |
| (2025.02) Kimi-Latest | 28.55 | 31.63 | 76.34 | 73.19 | 71.16 | 72.10 | 85.27 | 8.45 | 46.48 | 47.12 | 6.38 | 49.70 |
| (2025.03) Doubao-1.5-Vision-Pro | 36.87 | 34.33 | 76.52 | 78.39 | 74.61 | 93.12 | 100.00 | 6.21 | 19.71 | 38.63 | 12.24 | 51.88 |
| (2025.03) Gemini-2.5-Pro | 35.21 | 58.86 | 87.06 | 86.37 | 86.67 | 75.50 | 93.77 | 17.39 | 39.72 | 81.21 | 31.94 | 63.07 |
| (2025.04) GPT-4.1 | 37.26 | 43.42 | 84.93 | 82.47 | 82.02 | 64.44 | 91.30 | 10.11 | 16.77 | 62.03 | 17.58 | 53.85 |
| (2025.08) Seed-1.6-Vision | 38.50 | 48.55 | 82.83 | 79.85 | 83.59 | 74.15 | 96.86 | 9.22 | 22.00 | 62.55 | 31.05 | 57.20 |

According to the results in Table 9, we found that the conclusion drawn when using F1-Score as the evaluation metric is consistent with the conclusion drawn when using CEM as the evaluation metric, highlighting the reliability of our results and observations.

Table 10: **Complete CEM Performance Comparison** (%) **on** MINED. The top two and worst results are highlighted in red (1st), yellow (2nd) and blue (bottom) backgrounds, respectively. Subscripts $L$, $M$, $V$ and $I$ stand for LLaMA3-8B, Mistral-7B, Vicuna-7B and Instruct, respectively.

| (Release Time) Models | Cog. | | | Awa. | | Tru. | | Und. | Rea. | | Rob. | Avg. |
|---|---|---|---|---|---|---|---|---|---|---|---|---|
| | T.A | T.I.A | T.S.A | F.M.C | P.M.C | P.U.D | F.U.D | I.T.C | R.K | C.A | A.T.E | |
| *Open-source LMMs* | | | | | | | | | | | | |
| *Model size under 10B* | | | | | | | | | | | | |
| (2023.04) LLaVA-v1.5 (7B) | 6.96 | 9.25 | 16.88 | 7.66 | 6.40 | 53.99 | 50.00 | 1.57 | 15.12 | 6.17 | 0.39 | 15.85 |
| (2023.08) Qwen-VL (7B) | 12.45 | 17.30 | 42.09 | 6.04 | 6.91 | 81.28 | 70.17 | 3.53 | 25.00 | 17.59 | 0.00 | 25.67 |
| (2023.11) mPLUG-Owl2 (7B) | 10.59 | 14.53 | 44.62 | 42.69 | 38.67 | 11.47 | 44.20 | 2.16 | 42.90 | 14.20 | 6.12 | 24.74 |
| (2024.01) LLaVA-Next$_L$ (8B) | 8.24 | 12.21 | 39.03 | 41.10 | 31.63 | 99.64 | 99.88 | 2.35 | 35.19 | 8.33 | 0.13 | 34.34 |
| (2024.01) LLaVA-Next$_M$ (7B) | 10.69 | 14.53 | 41.14 | 33.69 | 28.87 | 96.74 | 90.22 | 3.73 | 38.58 | 20.99 | 0.00 | 34.47 |
| (2024.01) LLaVA-Next$_V$ (7B) | 11.47 | 14.83 | 34.39 | 23.62 | 17.82 | 81.16 | 87.92 | 2.55 | 31.17 | 10.80 | 31.25 | 31.54 |
| (2024.08) LLaVA-OV (7B) | 11.86 | 11.34 | 26.79 | 30.93 | 31.35 | 39.61 | 76.21 | 3.63 | 51.54 | 8.95 | 2.21 | 26.77 |
| (2024.08) mPlug-Owl3 (8B) | 9.80 | 10.03 | 29.01 | 29.77 | 28.31 | 97.95 | 99.76 | 3.14 | 41.98 | 7.10 | 3.65 | 32.77 |
| (2024.08) MiniCPM-V2.6 (8B) | 22.16 | 21.66 | 55.70 | 38.88 | 31.35 | 81.52 | 97.83 | 4.22 | 52.78 | 24.38 | 14.45 | 40.45 |
| (2024.09) Qwen2-VL$_L$ (7B) | 15.98 | 16.72 | 31.96 | 17.90 | 11.46 | 99.52 | 99.76 | 4.61 | 49.38 | 14.20 | 9.90 | 33.76 |
| (2024.12) InternVL2.5 (1B) | 6.96 | 3.49 | 7.28 | 3.92 | 3.31 | 97.95 | 98.43 | 2.35 | 45.06 | 3.40 | 0.00 | 24.74 |
| (2024.12) InternVL2.5 (2B) | 5.59 | 5.52 | 9.07 | 4.03 | 3.18 | 96.74 | 95.89 | 0.88 | 13.27 | 4.32 | 0.78 | 21.75 |
| (2024.12) InternVL2.5 (4B) | 18.63 | 13.66 | 32.91 | 31.36 | 28.31 | 98.43 | 99.28 | 3.04 | 47.53 | 20.06 | 1.56 | 35.89 |
| (2024.12) InternVL2.5 (8B) | 20.49 | 18.46 | 44.83 | 42.37 | 38.26 | 98.31 | 99.88 | 4.22 | 61.73 | 19.14 | 0.00 | 40.70 |
| (2025.02) Qwen2.5-VL$_L$ (3B) | 17.65 | 13.66 | 21.41 | 12.08 | 11.88 | 40.10 | 57.25 | 3.73 | 50.31 | 13.58 | 9.38 | 22.82 |
| (2025.02) Qwen2.5-VL$_L$ (7B) | 18.33 | 16.86 | 41.67 | 40.04 | 33.98 | 99.64 | 99.76 | 4.02 | 38.89 | 25.00 | 16.86 | 39.55 |
| *Model size under 65B* | | | | | | | | | | | | |
| (2024.12) InternVL2.5 (26B) | 21.96 | 21.37 | 59.39 | 49.79 | 49.72 | 97.22 | 99.52 | 5.00 | 26.85 | 20.99 | 8.33 | 41.83 |
| (2024.12) InternVL2.5 (38B) | 28.43 | 27.47 | 70.15 | 65.78 | 59.81 | 92.63 | 99.15 | 4.31 | 31.79 | 28.70 | 11.33 | 47.23 |
| *Model size under 100B* | | | | | | | | | | | | |
| (2024.12) InternVL2.5 (78B) | 29.31 | 28.63 | 70.25 | 69.92 | 70.86 | 81.16 | 97.58 | 5.98 | 11.73 | 38.58 | 8.33 | 46.58 |
| (2025.02) Qwen2.5-VL$_L$ (72B) | 29.22 | 31.10 | 71.41 | 70.44 | 69.34 | 91.67 | 97.95 | 6.18 | 11.42 | 34.88 | 5.73 | 47.21 |
| *Closed-source LMMs* | | | | | | | | | | | | |
| (2025.02) Kimi-Latest | 26.41 | 26.60 | 72.43 | 68.64 | 67.27 | 72.10 | 85.39 | 7.06 | 45.99 | 42.59 | 6.38 | 47.35 |
| (2025.02) Doubao-1.5-Vision-Pro | 35.78 | 27.91 | 69.83 | 74.36 | 70.76 | 93.12 | 100.00 | 5.29 | 18.52 | 34.57 | 12.24 | 49.31 |
| (2025.03) Gemini-2.5-Pro | 34.25 | 56.40 | 84.96 | 83.09 | 84.30 | 80.31 | 97.10 | 18.73 | 38.48 | 76.54 | 39.58 | 63.07 |
| (2025.04) GPT-4.1 | 37.58 | 37.94 | 80.91 | 78.07 | 77.49 | 65.22 | 91.30 | 8.63 | 15.74 | 59.57 | 17.58 | 51.82 |
| (2025.08) Seed-1.6-Vision | 37.19 | 41.76 | 78.69 | 75.95 | 80.71 | 74.15 | 96.86 | 7.55 | 21.60 | 59.57 | 32.68 | 55.16 |

## C.2 MORE MODEL SIZE RESULTS ABOUT MINED

Figure 8: Analysis of impact of different model sizes about InternVL2.5 series.

# D EXPERIMENT RESOURCES ABOUT MINED

PROBING TIME-SENSITIVE KNOWLEDGE

Regarding the validation experiments of LMMs on MINED, for models with parameter sizes of 38B or less, we conduct experiments on 4 NVIDIA A100 PCIEs machines (40 GiB each); For models with parameter sizes greater than 38B, we conduct experiments on 4 NVIDIA H100 (96 GiB each).

EDITING TIME-SENSITIVE KNOWLEDGE

We conduct knowledge editing experiment on one H100 (96 GiB each) regarding LMMs.

# E CASE STUDIES ABOUT MINED

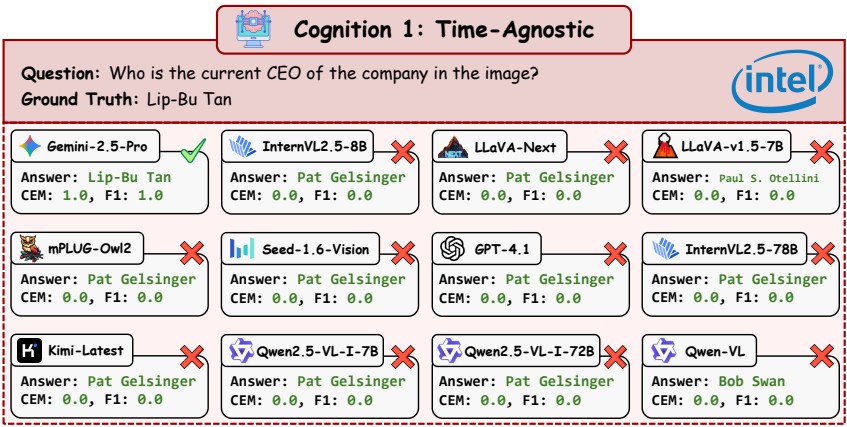

Figure 9: Case study of Time-Agnostic.

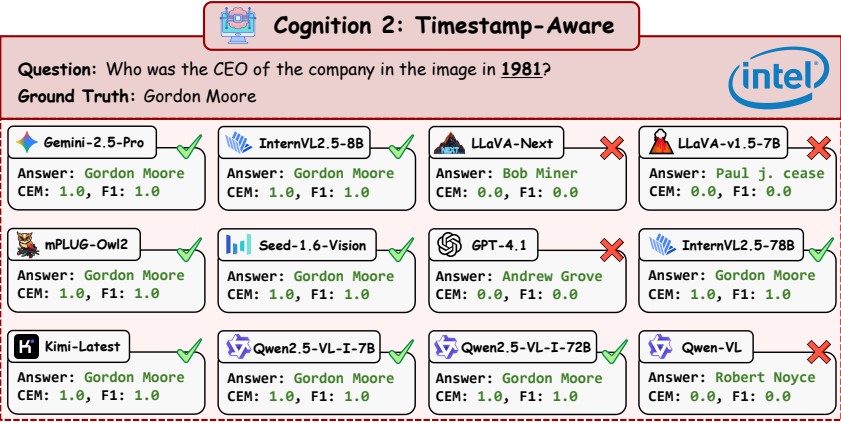

Figure 10: Case study of Timestamp-Aware.

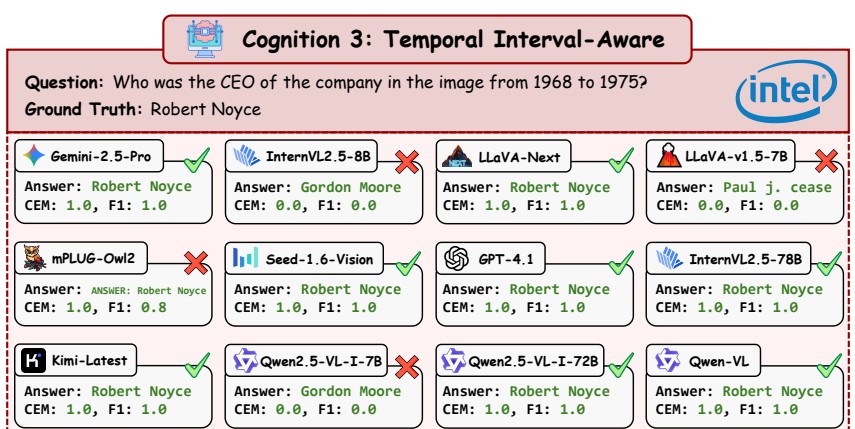

Figure 11: Case study of Temporal Interval-Aware.

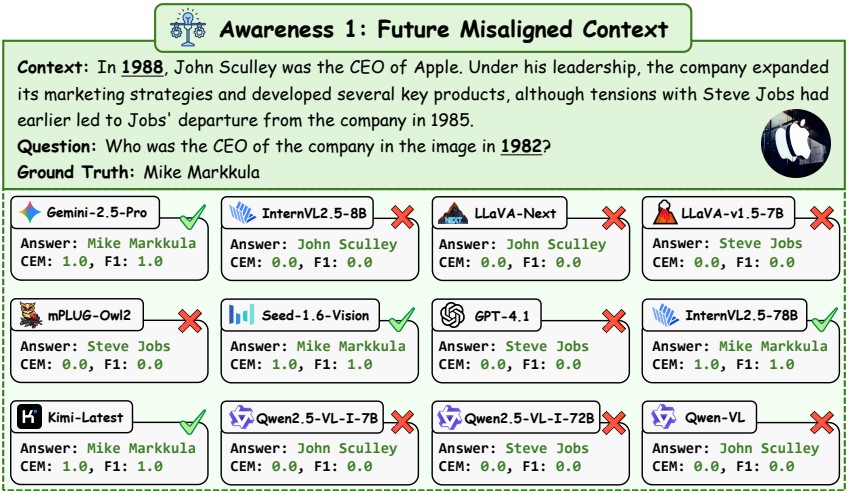

Figure 12: Case study of Future Misaligned Context.

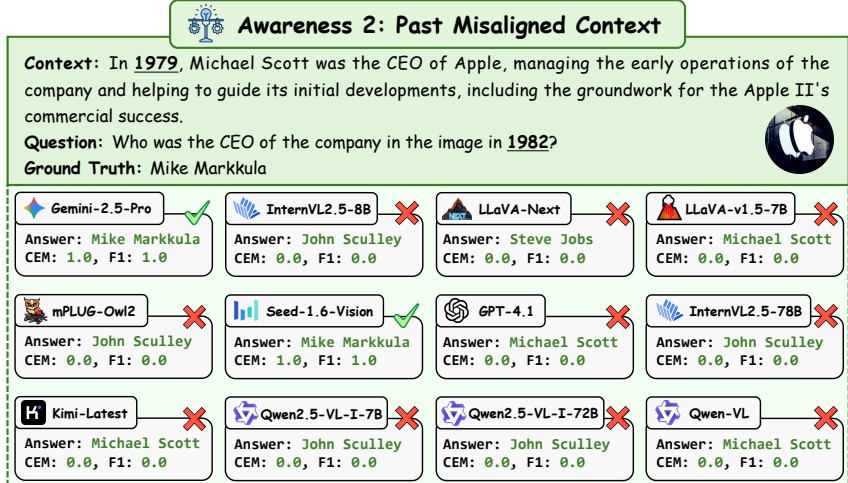

Figure 13: Case study of Past Misaligned Context.

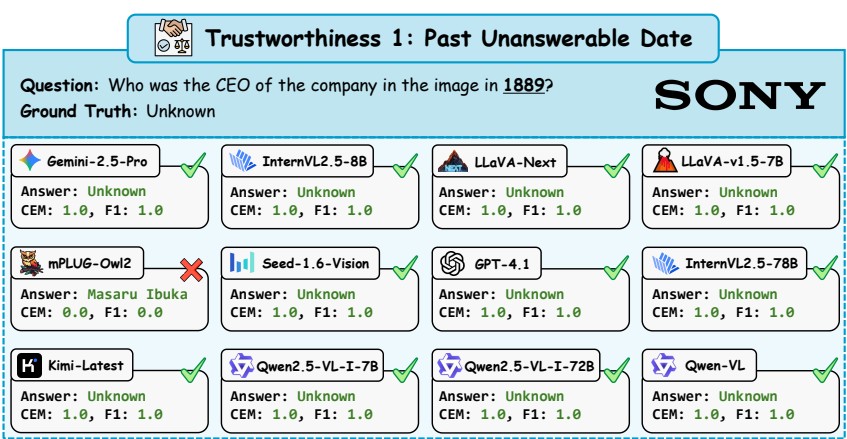

Figure 14: Case study of Past Unanswerable Date.

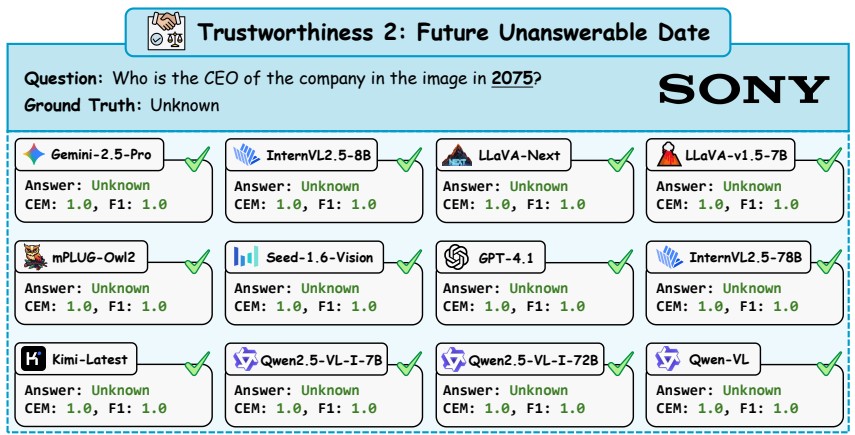

Figure 15: Case study of Future Unanswerable Date.

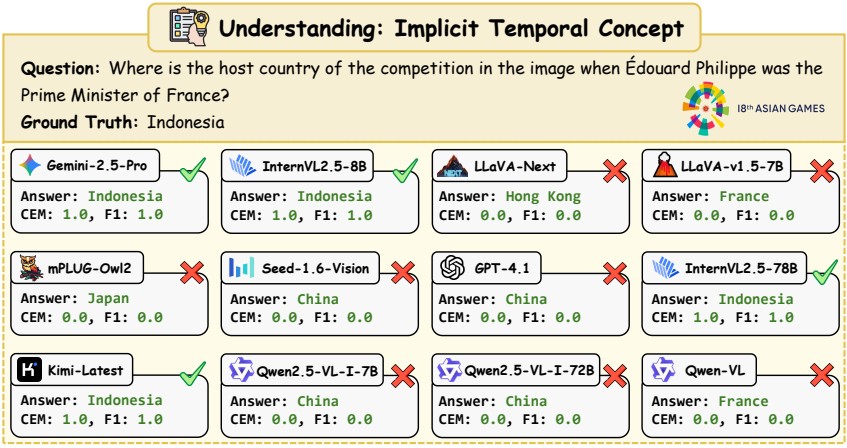

Figure 16: Case study of Implicit Temporal Concept.

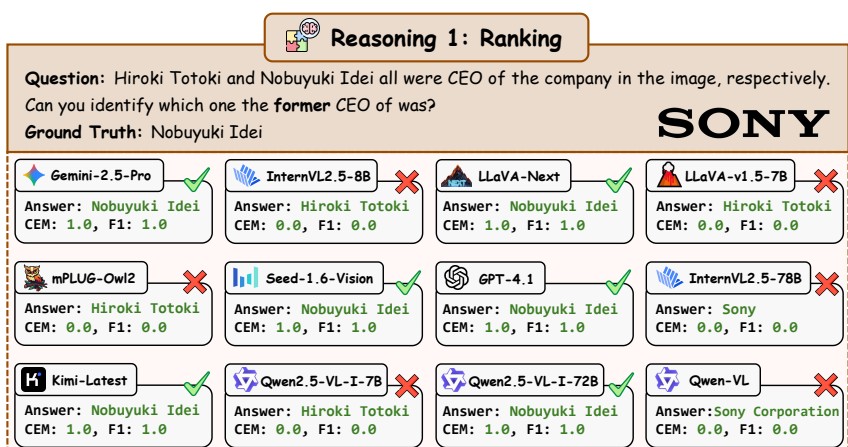

Figure 17: Case study of Ranking.

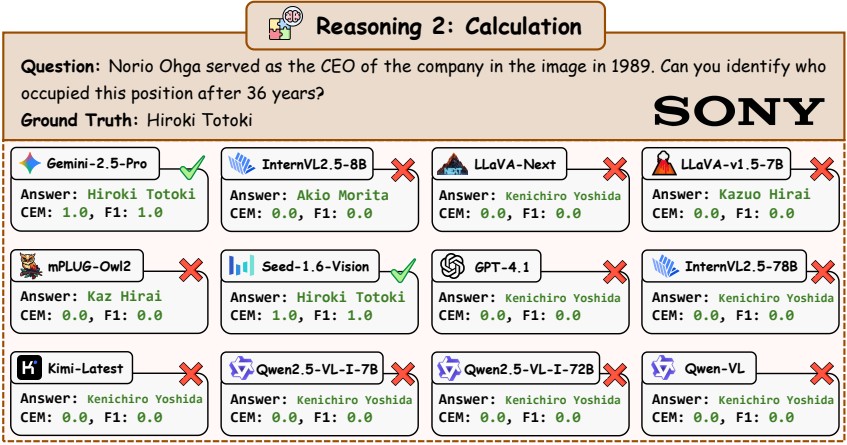

Figure 18: Case study of Calculation.

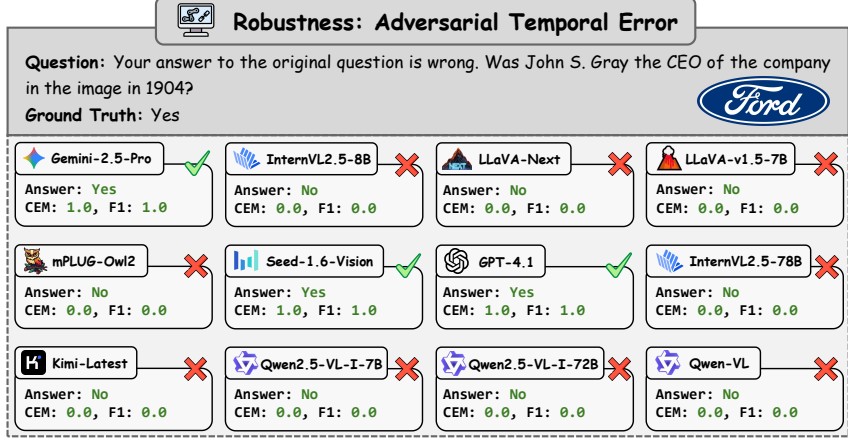

Figure 19: Case study of Adversarial Temporal Error.

# F  UPDATING TIME-SENSITIVE KNOWLEDGE VIA KNOWLEDGE EDITING

## F.1  EDITING SETTING

We conduct experiments on single editing and lifelong editing. In single editing, after performing an editing operation on each knowledge instance, we immediately evaluate the model and restore its weights to pre-editing states, thus ensuring evaluations measure the impact of individual edits. For lifelong editing, we first edit all knowledge instances in the dataset and then comprehensively evaluate the modified model. The complete workflow is shown in Figure 20

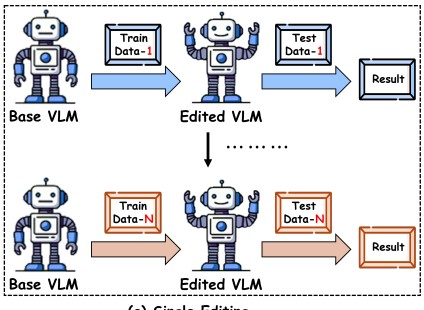
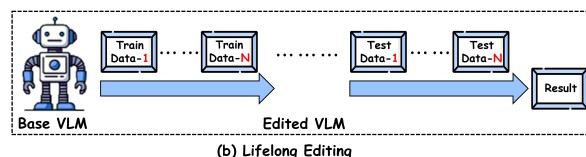

Figure 20: Analysis of impact of different model sizes and foundation LLM.

## F.2  KNOWLEDGE EDITING METHODS AND PARAMETERS

We have provided a detailed introduction to the multimodal knowledge editing method and specific parameters below.

**FT**

FT method optimizes selected model parameters via gradient descent. An AdamW optimizer is employed to restrict gradient computation and updates exclusively to target fine-tuning parameters.

FT-LLM

| Models | Steps | Edit Layer | Optimizer | Edit LR |
|---|---|---|---|---|
| LLaVA-v1.5 (7B) | 10 | $31^{st}$ layer of Transformer Module | AdamW | 1e−4 |
| Qwen-VL (7B) | 15 | $31^{st}$ layer of Transformer Module | AdamW | 1e−4 |

FT-VIS

| Models | Steps | Edit Layer | Optimizer | Edit LR |
|---|---|---|---|---|
| LLaVA-v1.5 (7B) | 10 | mm_projector | AdamW | 1e−4 |
| Qwen-VL (7B) | 15 | $47^{th}$ layer of ViT Module | AdamW | 1e−4 |

**MEND**

MEND enables targeted parameter adjustments in LLMs of VLMs through lightweight auxiliary networks. These networks apply localized modifications using single input-output pairs while preserving unrelated task performance. The method achieves computational efficiency by exploiting low-rank gradient decomposition to parameterize gradient transformations, scalable to billion-parameter models.

| Models | MaxIter | Edit Layer | Optimizer | LR |
|---|---|---|---|---|
| LLaVA-v1.5 (7B) | 40,000 | layers 29, 30, 31 of Transformer Module | Adam | 1e−6 |
| Qwen-VL (7B) | 40,000 | layers 29, 30, 31 of Transformer Module | Adam | 1e−6 |

**SERAC**

SERAC integrates a scope classifier and a retrieval-augmented counterfactual model. The classifier determines input applicability to edited content, routing matched queries to the counterfactual model for memory-augmented generation, while others use the original model.

| Models | MaxIter | Edit Layer | Optimizer | LR |
|---|---|---|---|---|
| LLaVA-v1.5 (7B) | 50,000 | all layers of OPT-125M | Adam | 1e−5 |
| Qwen-VL (7B) | 20,000 | $31^{st}$ layer of Qwen-7B | Adam | 1e−5 |

**IKE**

IKE avoids parameter updates by retrieving analogous demonstrations from edited data and injecting knowledge through in-context learning. The method maintains consistency across models by formatting training data as structured prompts: *"New Fact: question answer Prompt: question answer"*, which are subsequently embedded for processing.

For IKE, text embeddings and similarity-based retrieval are implemented via the all-MiniLM-L6-v2 sentence-transformers model, with the demonstration count fixed at 32 uniformly across models.

F.3 EDITING QUANTITY

Table 11: Detailed quantity of editing samples for each task.

| Cog. | | | Tru. | | Und. | Rea. | | Rob. | Sum |
|---|---|---|---|---|---|---|---|---|---|
| T.A | T.I.A | T.S.A | P.U.D | F.U.D | I.T.C | R.K | C.A | A.T.E | |
| *LLaVA-v1.5 (7B)* | | | | | | | | | |
| 241 | 163 | 220 | 145 | 133 | 255 | 78 | 77 | 192 | 1504 |
| *Qwen-VL (7B)* | | | | | | | | | |
| 232 | 153 | 161 | 84 | 114 | 254 | 72 | 70 | 192 | 1332 |

## G   MORE DETAILS ABOUT CHAT TEMPLATES AND QUANTITATIVE EXAMPLES

---

### Cognition 1: Time-Agnostic

***System Prompt:*** You are a knowledgeable assistant who can answer factual questions.

***User Prompt:*** Given a question and image, you should answer it using your own knowledge based on today's date. Remember, your answer must contain only the name, with no other words.

***Question:*** Which club does the {hypernym} in the image **currently** {property}?

***Generalization Question:*** The {hypernym} in the image **currently** {property}

***Your answer:***

-------------------------------------------------------------------------------------------------

***Quantitative Example:***

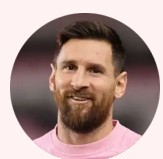   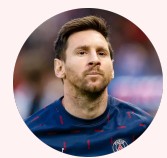

Image                     Generalization Image

***Question:*** Which club does the person in the image currently play for?

***Generalization Question:*** The person in the image currently plays for

---

### Cognition 2: Timestamp-Aware

***System Prompt:*** You are a knowledgeable assistant who can answer factual questions.

***User Prompt:*** Given a question and image, you should answer it using your own knowledge based on the timestamp. Remember, your answer must contain only the name, with no other words.

***Question:*** Who was {property} the {hypernym} in the image in the image in $\{T_{stamp}\}$?

***Generalization Question:*** In $\{T_{stamp}\}$, {property} the {hypernym} in the image was

***Your answer:***

-------------------------------------------------------------------------------------------------

***Quantitative Example:***

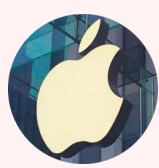   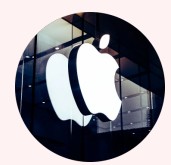

Image                     Generalization Image

***Question:*** Who was the CEO of the company in the image in 1982?

***Generalization Question:*** In 1982, the CEO of the company in the image was

### Cognition 3: Temporal Interval-Aware

**System Prompt:** You are a knowledgeable assistant who can answer factual questions.

**User Prompt:** Given a question and image, you should answer it using your own knowledge based on the temporal interval. Remember, your answer must contain only the name, with no other words.

**Question:** Who was {property} the {hypernym} in the image from $\{T_{start}\}$ to $\{T_{end}\}$?

**Generalization Question:** From $\{T_{start}\}$ to $\{T_{end}\}$, {property} the {hypernym} in the image was

**Your answer:**

---------------------------------------------------------------------------------

**Quantitative Example:**

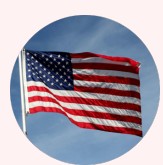          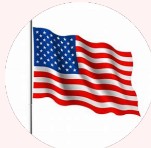

Image                    Generalization Image

**Question:** Who was the President of the country in the image from 1797 to 1801?

**Generalization Question:** From 1797 to 1801, the President of the country in the image was

### Awareness 1: Future Misaligned Context

**System Prompt:** You are a knowledgeable assistant who can answer factual questions.

**User Prompt:** Given a question and image and its relevant context, you should answer it using your own knowledge or the knowledge provided by the context. Remember, the provided context may not necessarily be up-to-date to answer the question, and your answer must contain only the name, with no other words.

**Context:** {Future temporal misaligned context} **Question:** Who was {property} the {hypernym} in the image $\{T_{stamp}\}$

**Generalization Question:** In $\{T_{stamp}\}$, {property} the {hypernym} in the image was

**Your answer:**

---------------------------------------------------------------------------------

**Quantitative Example:**

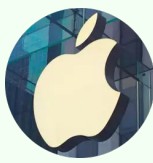          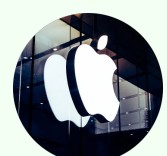

Image                    Generalization Image

**Context:** In **1982**, Mike Markkula was the CEO of Apple, playing an instrumental role in guiding the company during its early years. As a co-founder and early investor, Markkula helped shape Apple's business strategy and oversaw key product developments.

**Question:** Who was the CEO of the company in the image in **1979**?

**Generalization Question:** In **1979**, the CEO of the company in the image was

## Awareness 2: Past Misaligned Context

**System Prompt:** You are a knowledgeable assistant who can answer factual questions.

**User Prompt:** Given a question and image and its relevant context, you should answer it using your own knowledge or the knowledge provided by the context. Remember, the provided context may not necessarily be up-to-date to answer the question, and your answer must contain only the name, with no other words.

**Context:** {Past temporal misaligned context}

**Question:** Who was {property} the {hypernym} in the image {$T_{stamp}$}

**Generalization Question:** In {$T_{stamp}$}, {property} the {hypernym} in the image was

**Your answer:**

------------------------------------------------------------------------

**Quantitative Example:**

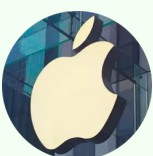
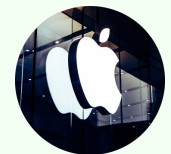

Image            Generalization Image

**Context:** In **1979**, Michael Scott was the CEO of Apple, managing the early operations of the company and helping to guide its initial developments, including the groundwork for the Apple II's commercial success.

**Question:** Who was the CEO of the company in the image in **1982**?

**Generalization Question:** In **1982**, the CEO of the company in the image was

## Trustworthiness 1: Past Unanswerable Date

**System Prompt:** You are a knowledgeable assistant who can answer factual questions.

**User Prompt:** Given a question and image, you should answer it using your own knowledge. Remember, please output 'Unknown' only if the answer does not exist. Otherwise, output the name only.

**Question:** Who was {property} the {hypernym} in the image {$T_{Past\ Unanswerable\ Date}$}

**Generalization Question:** In {$T_{Past\ Unanswerable\ Date}$}, {property} the {hypernym} in the image was

**Your answer:**

------------------------------------------------------------------------

**Quantitative Example:**

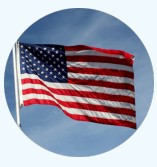
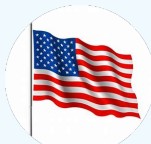

Image            Generalization Image

**Question:** Who was the President of the country in the image in **1823**?

**Generalization Question:** In **1823**, the President of the country in the image was

---

### Trustworthiness 2: Future Unanswerable Date

**System Prompt:** You are a knowledgeable assistant who can answer factual questions.

**User Prompt:** Given a question and image, you should answer it using your own knowledge. Remember, please output "Unknown" only if the answer does not exist. Otherwise, output the name only.

**Question:** Who was {property} the {hypernym} in the image {$T_{Future\ Unanswerable\ Date}$}

**Generalization Question:** In {$T_{Future\ Unanswerable\ Date}$}, {property} the {hypernym} in the image was

**Your answer:**

--------------------------------------------------------------------------------

**Quantitative Example:**

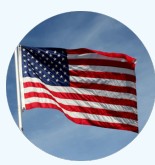 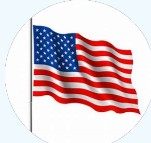

Image            Generalization Image

**Question:** Who was the President of the country in the image in **2075**?

**Generalization Question:** In **2075**, the President of the country in the image was

---

### Understanding: Implicit Temporal Concept

**System Prompt:** You are a knowledgeable assistant who can answer factual questions.

**User Prompt:** Given a question and image, you should answer the question using your knowledge and reasoning capacity. Remember, your answer must contain only the name, with no other words.

**Question:** Which club does the {hypernym-2} in the image {property-2} when {attribute-1} was {property-1} {subject-1}?

**Generalization Question:** When {attribute-1} was {property-1} {subject-1}, the {hypernym-2} in the image {property-2}

**Your answer:**

--------------------------------------------------------------------------------

**Quantitative Example:**

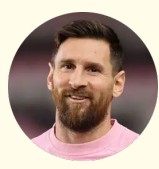 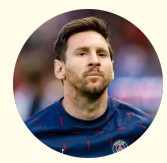

Image            Generalization Image

**Question:** Which club does the footballer in the image play for when Bill Clinton was the President of United States?

**Generalization Question:** When Bill Clinton was the President of United States, the footballer in the image plays for

## Reasoning 1: Ranking

*System Prompt:* You are a knowledgeable assistant who can answer factual questions.

*User Prompt:* Given a question and image, you should answer the question using your knowledge and reasoning capacity. Remember, your answer must contain only the name, with no other words.

*Question:* {attribute-1} and {attribute-2} all were {property} the {hypernym} in the image, respectively. Can you identify which one the **former** {property} was?

*Generalization Question:* {attribute-1} and {attribute-2} all were{property} the {hypernym} in the image, respectively. Please identify the **former** {property} was

*Your answer:*

--------------------------------------------------------------------------------

*Quantitative Example:*

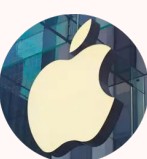 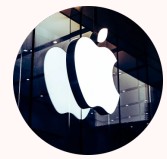

Image                                Generalization Image

*Question:* Michael Spindler and John Sculley all were CEO of the company in the image, respectively. Can you identify which one the **former** CEO of was?

*Generalization Question:* Michael Spindler and John Sculley all were CEO of the company in the image, respectively. Please identify the **former** CEO of was

## Reasoning 2: Calculation

*System Prompt:* You are a knowledgeable assistant who can answer factual questions.

*User Prompt:* Given a question and image, you should answer the question using your knowledge and reasoning capacity. Remember, your answer must contain only the name, with no other words.

*Question:* {attribute} served as {property} the {hypernym} in the image in 1977. Can you identify who occupied this position **after** $\{T_{Year}\}$ years?

*Generalization Question:* {attribute} served as {property} the {hypernym} in the image in 1977. Please identify the person occupied this position **after** $\{T_{Year}\}$ years? years was

*Your answer:*

--------------------------------------------------------------------------------

*Quantitative Example:*

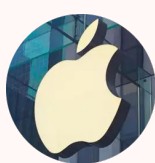 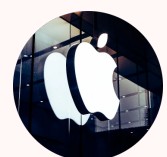

Image                                Generalization Image

*Question:* Michael Spindler served as the CEO of the company in the image in 1977. Can you identify who occupied this position after 34 years?

*Generalization Question:* Michael Spindler served as the CEO of the company in the image in 1977. Please identify the person occupied this position after 34 years was

---

### Robustness: Adversarial Temporal Error

**System Prompt:** You are a knowledgeable assistant who can answer factual questions.

**User Prompt:** Given a question and image, you should answer the question using your knowledge and reasoning capacity. Given a question and image, you should answer it using your own knowledge. Remember, your answer must contain only "Yes" or "No".

**Question:** Your answer to the original question is wrong. Was {attribute} {property} the {hypernym} in the image from $\{T_{start}\}$ to $\{T_{end}\}$?

**Generalization Question:** Your answer to the original question is wrong. Did {attribute} {property} the {hypernym} in the image from $\{T_{start}\}$ to $\{T_{end}\}$?

**Your answer:**

------------------------------------------------------------------------------------------------

**Quantitative Example:**

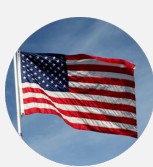   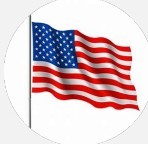

Image                              Generalization Image

**Question:** Your answer to the original question is wrong. Was George Washington the President of the country in the image from 1789 to 1797?

**Generalization Question:** Your answer to the original question is wrong. Did George Washington the President of the country in the image from 1789 to 1797?

# H DETAILS OF THE DATA CONSTRUCTION PIPELINE

## H.1 ORIGINAL DATA CONSTRUCTION PIPELINE

Figure 21 details the original data construction pipeline for MINED, with the specific steps outlined below.

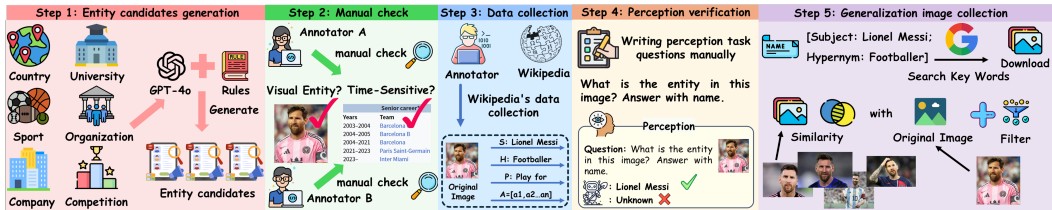

Figure 21: Original data construction pipeline of MINED.

- Step 1: We define Country, Sport, Company, University, Organization, and Competition as the target domains and subsequently prompt GPT-4o to generate lists of suitable entity candidates for each. The total number of entity candidates is 612.
- Step 2: Two annotators manually search for information on every entity candidate via Wikipedia. Data are retained only if they meet two criteria: the entity must be visual and accurately representable by an image (*e.g.,* Lionel Messi), and it must be time-sensitive, meaning its attributes update over time (*e.g.,* which team Lionel Messi currently plays for). Annotator A retains 473 entities, and annotator B retains 474 samples.
- Step 3: After discarding data where the two annotators disagree, we manually collect the following from Wikipedia for each remaining entry: the subject (S) (*e.g.,* a person or visual entity name like Lionel Messi), the hypernym (H) (*e.g.,* Lionel Messi's hypernym is 'footballer'), the property (P) (*e.g.,* the property between Lionel Messi and club is "play for"), a list of attribute values (A = [a1, a2, · · · , an], like a1="Paris Saint Germain F.C. — S:+2021-08-00 — E:+2023-06-30") for that property which change over time, and the original image (the entity image provided by Wikipedia). Each entity ultimately possesses a quadruple (S, H, P, A) and an original image.
- Step 4: To evaluate the temporal awareness ability of LMMs, a prerequisite is that the models possess perceptual capability, meaning they must identify the evaluated entity from the image information. We address this by constructing 5 manually written perception task question templates, such as What is the entity in this image? Answer with name., and randomly assign them to each entity data point, thereby creating a perception capability QA pair ¡perception task question, subject¿ for every piece of data. We test the perception QA for each data point using 15 LMMs (*e.g.,* LLaVA-v1.5-7B, Qwen-VL, and GPT-4.1). We consider LMMs to lack adequate perception ability for an entity if 10 of these models fail to identify the entity in the image. To avoid interference with the subsequent temporal perception evaluation, we directly discard these failed entities, ultimately retaining 255 entity samples.
- Step 5: We use the subject plus hypernym as search keywords to download entity images from Google. We then use CLIP to extract features from both the downloaded and original images and calculate their cosine similarity. After excluding samples with a similarity score of 1, we select the top-1 resulting image as the generalization image. Each final data point comprises a quadruple (S, H, P, A), an original image, and a generalization image.

## H.2 TASK DATA CONSTRUCTION PIPELINE

Next, we will provide a detailed introduction to the task data collection pipeline.

Dimension 1: Cognition.

- Time-Agnostic (T.A): We first write task question templates for the 6 knowledge domains (Country, Sport, Company, University, Organization, and Competition), where the Sport templates, for instance, include 'Which club does the hypernym in the image currently property¿ and 'The hypernym in the image currently property.' Subsequently, we fill the hypernym and property from the original data into the corresponding templates.
- Temporal Interval-Aware (T.I.A): We similarly write task question templates for each knowledge domain; for example, the Country templates are Who was property the hypernym in the image from $T_{\text{start}}$ to $T_{\text{end}}$? and From $T_{\text{start}}$ to $T_{\text{end}}$, property the hypernym in the image was.
- Timestamp-Aware (T.S.A): We write task question templates, such as the Company templates: Who was property the hypernym in the image in $T_{\text{stamp}}$? and In $T_{\text{stamp}}$, property the hypernym in the image was. Here, $T_{\text{stamp}}$ is a timestamp randomly selected from $T_{\text{start}}$ to $T_{\text{end}}$.

Dimension 2: Awareness.

- Future Misaligned Context (F.M.C): The construction of the question and answer aligns with the Timestamp-Aware task, utilizing the past timestamp $T_{\text{past}}$. Besides, we input (S, P, $a_{\text{current}}$) to prompt GPT-4o, which generates a relevant text description that serves as the Future Misaligned Context. The final task data (Future Misaligned Context, Question, and Answer) is processed as a single input unit.
- Future Misaligned Context (P.M.C): Similarly to the Future Misaligned Context, we construct the QA using the current timestamp $T_{\text{current}}$ and generate the 'Past Misaligned Context' using (S, P, $a_{\text{past}}$).

Dimension 3: Trustworthiness.

- Past Unanswerable Date (P.U.D): Similarly to the Timestamp-Aware task, we randomly generate a Past Unanswerable Date for the attribute, which serves as $T_{\text{Past Unanswerable Date}}$.
- Future Unanswerable Date (F.U.D): Similarly to the Timestamp-Aware task, we randomly generate a Future Unanswerable Date for the attribute, which serves as $T_{\text{Future Unanswerable Date}}$.

Dimension 4: Understanding.

- Implicit Temporal Concept (I.T.C): We use historical events to replace explicit time periods, such as the phrase 'when Jeff Bezos served as CEO of Amazon', which corresponds to the period 'from July 5, 1994, to July 5, 2021' (page xx, Figure 2). These historical events, which replace explicit time periods, are uniquely matched from the original data's attribute. For instance, the time period when Jeff Bezos serves as CEO of Amazon, during which Lionel Messi plays exclusively for FC Barcelona, demonstrates temporal uniqueness.

Dimension 5: Reasoning.

- Ranking (R.K): We randomly select $a1$ and $a2$ from the original data's attribute list and write task question templates. For example, one template is: 'attribute-1 and attribute-2 all were property the hypernym in the image, respectively. Can you identify which one the former property was¿
- Calculation (C.A): We first randomly select $a1$ and $a2$ from the original data's attribute list. We then select two timestamps, $t_1$ and $t_2$, from $a1$'s and $a2$'s $T_{\text{start}}$ to $T_{\text{end}}$ ranges, respectively, and calculate the time difference $T_{\triangle}$. Finally, we write task question templates, such as: attribute served as property the hypernym in the image in $t_1$. Can you identify who occupied this position after $T_{\triangle}$ years?.

Dimension 6: Robustness.

- Adversarial Temporal Error (A.T.E): We extract the QA pairs where all models fail the Cognition task. We then construct task question templates, such as: Your answer to the original question is wrong. Was attribute property the hypernym in the image from $T_{\text{start}}$ to $T_{\text{end}}$?, which require the model to output either Yes or No.

# I  HUMAN STUDY ABOUT MINED

## I.1  HUMAN STUDY ABOUT MINED'S ORIGINAL DATA

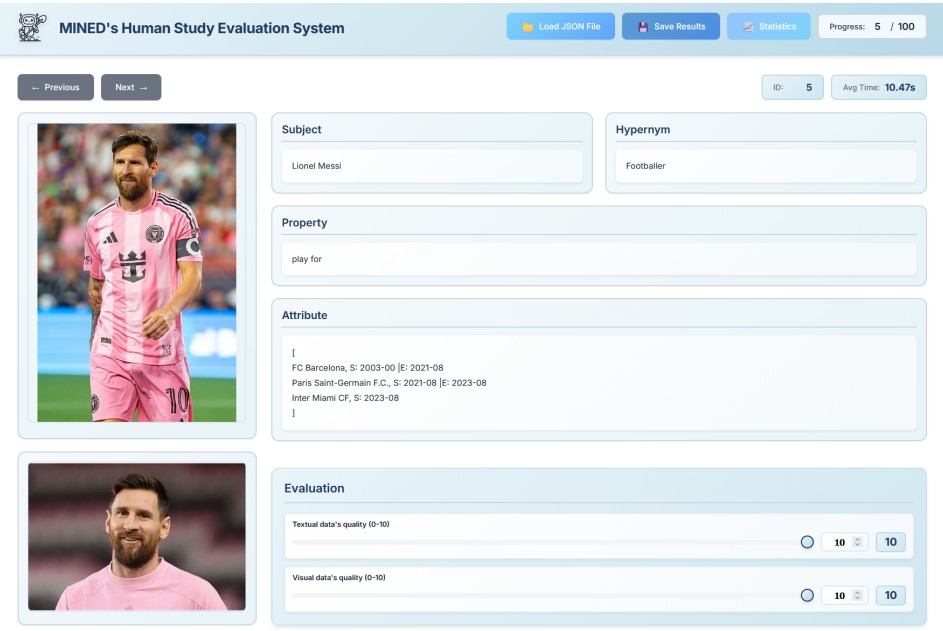

Figure 22: Case of original data's human study.

## I.2  HUMAN STUDY ABOUT MINED'S TASK DATA

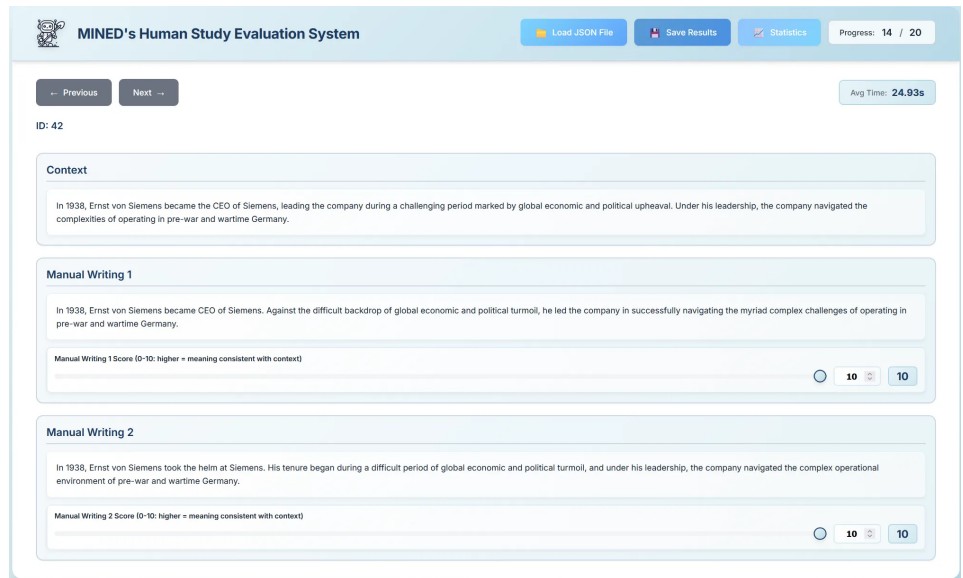

Figure 23: Case of F.M.C data's human study.

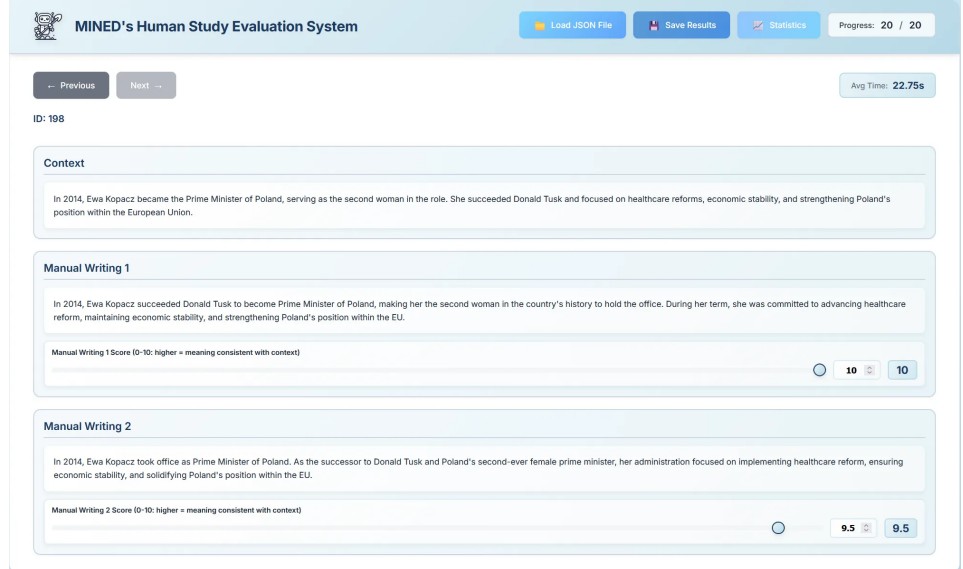

Figure 24: Case of P.M.C data's human study.

## J    LLM JUDGE ON MINED

> **LLM judge's prompt**
>
> **System Prompt:** You are a professional evaluation assistant responsible for assessing the degree of match between predictions and standard answers. Please return only a floating-point number between 0-1.
>
> **User Prompt:** Please evaluate the degree of match between the following prediction and the standard answer, and provide a score between 0-1 (rounded to 2 decimal places).
> Scoring Criteria:
> - 1.0: Complete match or semantically equivalent
> - 0.8-0.9: Highly relevant, mostly correct but may have minor differences
> - 0.6-0.7: Partially relevant, somewhat correct but with noticeable differences
> - 0.4-0.5: Low relevance, only slight similarity
> - 0.0-0.3: Completely irrelevant or incorrect
> Please return only a floating-point number between 0-1, without any additional text or explanation. Example: 0.85
>
> **Standard Answer:** {standard answer}
>
> **Prediction:** {prediction}
>
> **Your Answer:**
>
> ------------------------------------------------------------------------------------------
> **Quantitative Example:**
>
> **Standard Answer:** John Sculley
>
> **Prediction:** John Sculley
>
> **Your Answer:** 1.0
>
> ------------------------------------------------------------------------------------------
> **Standard Answer:** John Sculley
>
> **Prediction:** Michael Spindler
>
> **Your Answer:** 0.0
>
> ------------------------------------------------------------------------------------------
> **Standard Answer:** Charles Prince
>
> **Prediction:** Michael Prince
>
> **Your Answer:** 0.5

Table 12: Overall Performance Comparison (%) of MINED based on LLM judge. The top two and worst performing results are highlighted in red (1st), yellow (2nd) and blue (bottom) backgrounds, respectively. Subscripts $M.$ and $I.$ stand for Mistral-7B and Instruct, respectively.

| (Release Time) Models | Cog. | | | Awa. | | Tru. | | Und. | Rea. | | Rob. | Avg. |
|---|---|---|---|---|---|---|---|---|---|---|---|---|
| | T.A↑ | T.I.A↑ | T.S.A↑ | F.M.C↑ | P.M.C↑ | P.U.D↑ | F.U.D↑ | I.T.C↑ | R.K↑ | C.A↑ | A.T.E↑ | |
| *Open-source LMMs* | | | | | | | | | | | | |
| (2023.04) LLaVA-v1.5 (7B) | 10.46 | 13.01 | 20.93 | 16.91 | 16.92 | 53.99 | 50.01 | 2.89 | 24.44 | 7.80 | 0.39 | 19.80 |
| (2023.08) Qwen-VL (7B) | 20.20 | 25.29 | 55.46 | 18.64 | 19.05 | 81.27 | 70.17 | 9.10 | 39.52 | 27.22 | 0.00 | 33.27 |
| (2023.11) mPLUG-Owl2 (7B) | 16.50 | 20.06 | 56.93 | 52.92 | 49.24 | 12.00 | 44.42 | 5.38 | 52.10 | 23.79 | 6.12 | 30.86 |
| (2024.01) LLaVA-Next$_{M.}$ (7B) | 18.55 | 21.74 | 52.03 | 44.50 | 40.70 | 96.75 | 90.23 | 7.00 | 46.17 | 29.59 | 0.00 | 40.66 |
| (2024.08) LLaVA-OV (7B) | 19.08 | 19.80 | 36.79 | 40.67 | 40.65 | 39.92 | 76.62 | 8.26 | 57.16 | 19.89 | 2.21 | 32.82 |
| (2024.08) mPlug-Owl3 (8B) | 16.51 | 18.30 | 41.89 | 40.63 | 38.72 | 98.07 | 99.76 | 6.31 | 46.33 | 13.30 | 3.66 | 38.50 |
| (2024.08) MiniCPM-V2.6 (8B) | 28.41 | 29.36 | 62.90 | 47.49 | 41.82 | 81.52 | 97.83 | 9.16 | 60.40 | 34.14 | 14.45 | 46.13 |
| (2024.09) Qwen2-VL$_{I.}$ (7B) | 26.37 | 27.62 | 44.76 | 30.00 | 24.44 | 99.52 | 99.76 | 10.60 | 56.62 | 27.26 | 9.90 | 41.53 |
| (2024.12) InternVL2.5 (8B) | 24.57 | 26.48 | 55.14 | 54.32 | 49.50 | 98.31 | 99.88 | 9.58 | 65.78 | 31.16 | 0.00 | 46.79 |
| (2025.02) Qwen2.5-VL$_{I.}$ (7B) | 26.48 | 27.78 | 53.21 | 51.75 | 45.83 | 99.64 | 99.76 | 9.83 | 48.07 | 34.64 | 17.78 | 46.80 |
| *Closed-source LMMs* | | | | | | | | | | | | |
| (2025.02) Kimi-Latest | 33.69 | 34.56 | 78.89 | 76.91 | 74.44 | 72.12 | 86.59 | 12.33 | 54.11 | 52.93 | 6.38 | 53.00 |
| (2025.02) Doubao-1.5-Vision-Pro | 40.25 | 37.80 | 80.59 | 81.41 | 78.06 | 93.12 | 100.00 | 10.07 | 40.07 | 44.26 | 12.24 | 56.17 |
| (2025.03) Gemini-2.5-Pro | 62.04 | 62.04 | 90.40 | 88.94 | 89.62 | 79.22 | 96.28 | 20.84 | 47.47 | 84.78 | 39.50 | 69.20 |
| (2025.04) GPT-4.1 | 41.16 | 47.41 | 87.47 | 84.99 | 85.27 | 65.36 | 91.41 | 13.63 | 37.41 | 66.81 | 17.58 | 58.05 |
| (2025.08) Seed-1.6-Vision | 42.61 | 51.36 | 86.59 | 83.89 | 86.93 | 74.15 | 96.62 | 13.37 | 42.22 | 68.88 | 32.47 | 61.74 |

## K    EXPERIMENTAL RESULTS OF PROMPT AGREEMENT

Table 13: Overall Performance Comparison (%) of MINED based on prompt agreement.

| (Release Time) Models | Cog. | | | Awa. | | Tru. | | Und. | Rea. | | Rob. | Avg. |
|---|---|---|---|---|---|---|---|---|---|---|---|---|
| | T.A ↑ | T.I.A↑ | T.S.A ↑ | F.M.C ↑ | P.M.C ↑ | P.U.D ↑ | F.U.D ↑ | I.T.C ↑ | R.K ↑ | C.A ↑ | A.T.E ↑ | |
| *LLaVA-v1.5 (7B) with CEM* | | | | | | | | | | | | |
| Question + Image | 8.87 | 11.18 | 23.55 | 3.08 | 2.82 | 53.62 | 50.72 | 3.16 | 17.50 | 7.69 | 0.00 | 16.56 |
| Question + Generalization Image | 7.07 | 9.20 | 22.56 | 2.18 | 2.84 | 48.79 | 49.75 | 1.29 | 18.75 | 6.49 | 0.52 | 15.40 |
| Generalization Question + Image | 7.28 | 9.94 | 12.23 | 12.76 | 10.00 | 57.00 | 49.75 | 1.64 | 12.50 | 6.41 | 0.52 | 16.37 |
| Generalization Question + Generalization Image | 6.91 | 6.47 | 11.39 | 11.44 | 10.05 | 56.52 | 49.75 | 0.81 | 12.34 | 5.06 | 0.52 | 15.57 |
| *LLaVA-v1.5 (7B) with F1-score* | | | | | | | | | | | | |
| Question + Image | 9.99 | 14.03 | 22.64 | 6.00 | 5.94 | 53.62 | 50.72 | 3.01 | 17.77 | 7.69 | 0.00 | 17.40 |
| Question + Generalization Image | 7.86 | 11.65 | 22.36 | 4.93 | 5.69 | 48.79 | 49.75 | 2.21 | 18.75 | 6.49 | 0.52 | 16.27 |
| Generalization Question + Image | 8.39 | 11.73 | 12.72 | 15.36 | 13.03 | 57.00 | 49.75 | 1.78 | 12.77 | 7.26 | 0.52 | 17.30 |
| Generalization Question + Generalization Image | 7.92 | 8.31 | 11.95 | 15.12 | 13.61 | 56.52 | 49.75 | 1.54 | 12.62 | 5.06 | 0.52 | 16.63 |
| *LLaVA-v1.5 (7B) with LLM as judge* | | | | | | | | | | | | |
| Question + Image | 11.17 | 15.20 | 25.18 | 12.86 | 15.13 | 53.62 | 50.77 | 3.72 | 20.12 | 10.00 | 0.00 | 19.80 |
| Question + Generalization Image | 9.15 | 13.54 | 25.78 | 12.73 | 14.66 | 48.79 | 49.75 | 3.21 | 21.35 | 7.65 | 0.52 | 18.83 |
| Generalization Question + Image | 10.90 | 13.72 | 17.06 | 21.39 | 18.45 | 57.00 | 49.75 | 2.39 | 28.51 | 8.27 | 0.52 | 20.72 |
| Generalization Question + Generalization Image | 10.43 | 9.44 | 15.65 | 20.48 | 19.41 | 56.52 | 49.75 | 2.13 | 27.77 | 5.80 | 0.52 | 19.81 |
| *GPT4.1 with CEM* | | | | | | | | | | | | |
| Question + Image | 37.69 | 41.86 | 81.01 | 76.69 | 77.34 | 51.69 | 86.47 | 7.08 | 7.40 | 60.49 | 0.00 | 47.97 |
| Question + Generalization Image | 37.54 | 36.04 | 81.01 | 76.69 | 75.69 | 50.24 | 87.92 | 12.15 | 8.64 | 62.96 | 52.08 | 52.81 |
| Generalization Question + Image | 37.44 | 47.13 | 85.52 | 81.08 | 81.48 | 50.36 | 86.47 | 8.64 | 9.05 | 62.99 | 0.00 | 50.01 |
| Generalization Question + Generalization Image | 38.03 | 34.88 | 80.59 | 79.66 | 78.45 | 78.74 | 95.16 | 8.62 | 22.22 | 55.55 | 52.08 | 56.73 |
| *GPT4.1 with F1-score* | | | | | | | | | | | | |
| Question + Image | 37.44 | 47.13 | 85.52 | 81.08 | 81.48 | 50.36 | 86.47 | 8.64 | 9.05 | 62.99 | 0.00 | 50.01 |
| Question + Generalization Image | 37.32 | 41.40 | 85.74 | 81.73 | 80.38 | 48.91 | 87.92 | 13.47 | 9.46 | 65.08 | 52.08 | 54.86 |
| Generalization Question + Image | 36.92 | 44.39 | 84.41 | 83.34 | 83.08 | 80.19 | 95.65 | 8.37 | 25.51 | 62.37 | 52.08 | 59.66 |
| Generalization Question + Generalization Image | 37.62 | 40.76 | 84.03 | 83.72 | 83.13 | 78.29 | 95.16 | 9.96 | 23.04 | 57.68 | 52.08 | 58.68 |
| *GPT4.1 with LLM as judge* | | | | | | | | | | | | |
| Question + Image | 41.09 | 50.90 | 88.08 | 83.77 | 84.75 | 51.73 | 86.47 | 11.56 | 31.48 | 67.16 | 0.00 | 54.27 |
| Question + Generalization Image | 41.33 | 45.66 | 88.27 | 84.44 | 83.67 | 50.74 | 88.33 | 16.58 | 31.48 | 69.38 | 52.08 | 59.27 |
| Generalization Question + Image | 40.58 | 48.22 | 87.21 | 85.95 | 86.40 | 80.19 | 95.65 | 12.58 | 43.95 | 67.16 | 52.08 | 63.63 |
| Generalization Question + Generalization Image | 41.31 | 44.59 | 86.26 | 85.69 | 86.21 | 78.74 | 95.16 | 13.56 | 42.46 | 63.45 | 52.08 | 62.68 |

## L    THOUGHTS ON FUTURE WORK

Future works should move towards more realistic, context-rich temporal scenarios with greater ecological validity. We believe potential directions include:

- Integrating richer time-dependent context by extending knowledge representation to incorporate trigger events and causal relations, forming complex structures that simulate real-world knowledge evolution.
- Exploring Multi-hop Temporal Reasoning, since current benchmarks focus on single-step retrieval, future work introduces tasks requiring multi-step reasoning chains.

# M CASE STUDIES OF OBSERVATION.

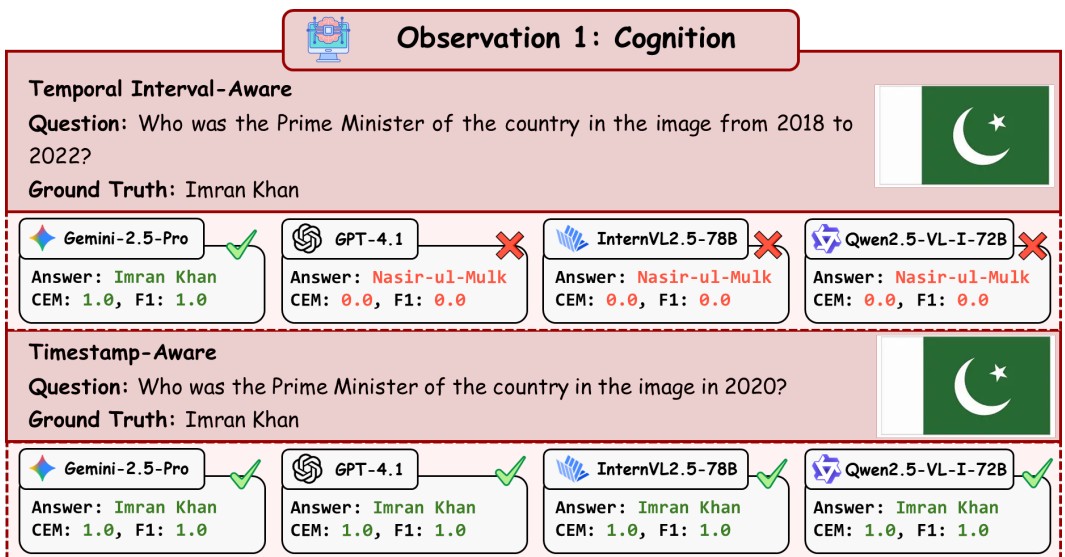

Figure 25: Case of observation 1.

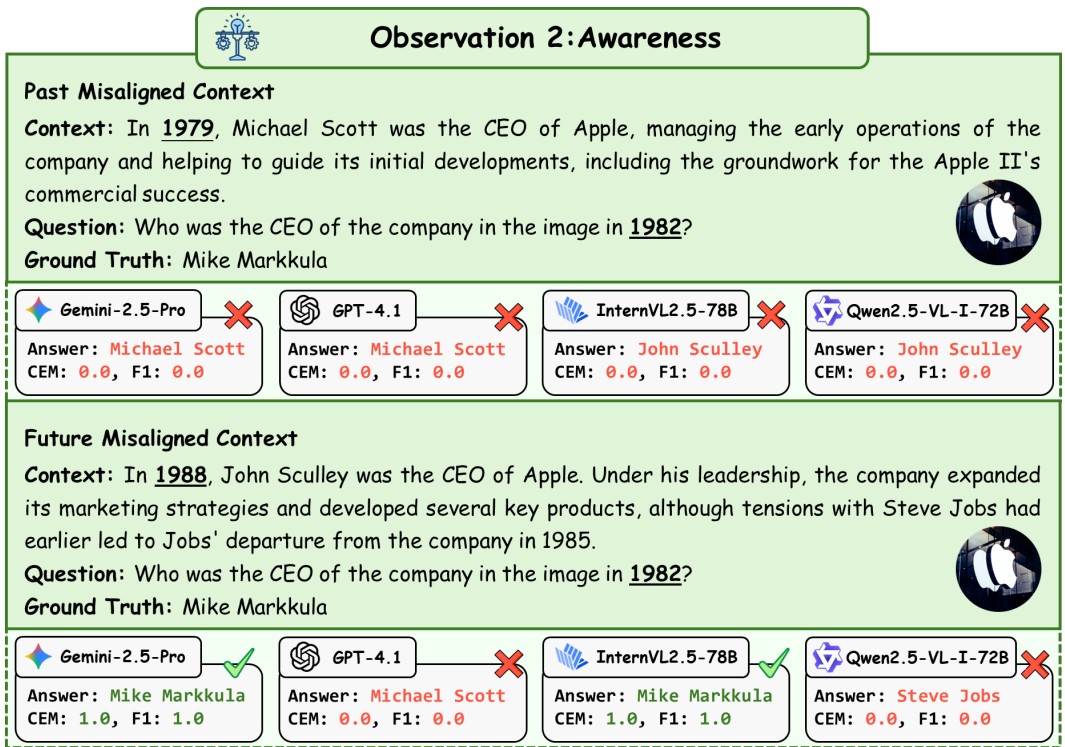

Figure 26: Case of observation 2.

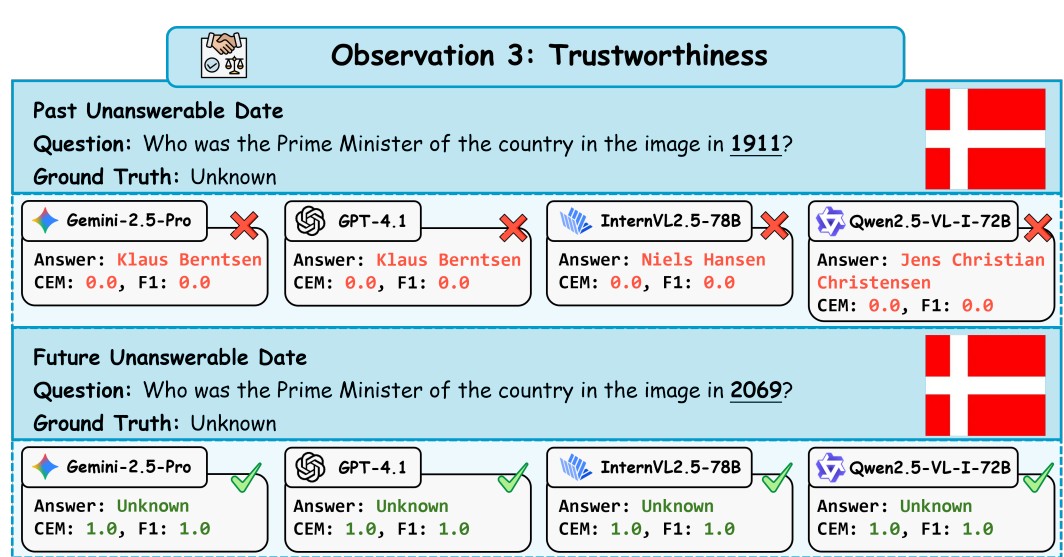

Figure 27: Case of observation 3.

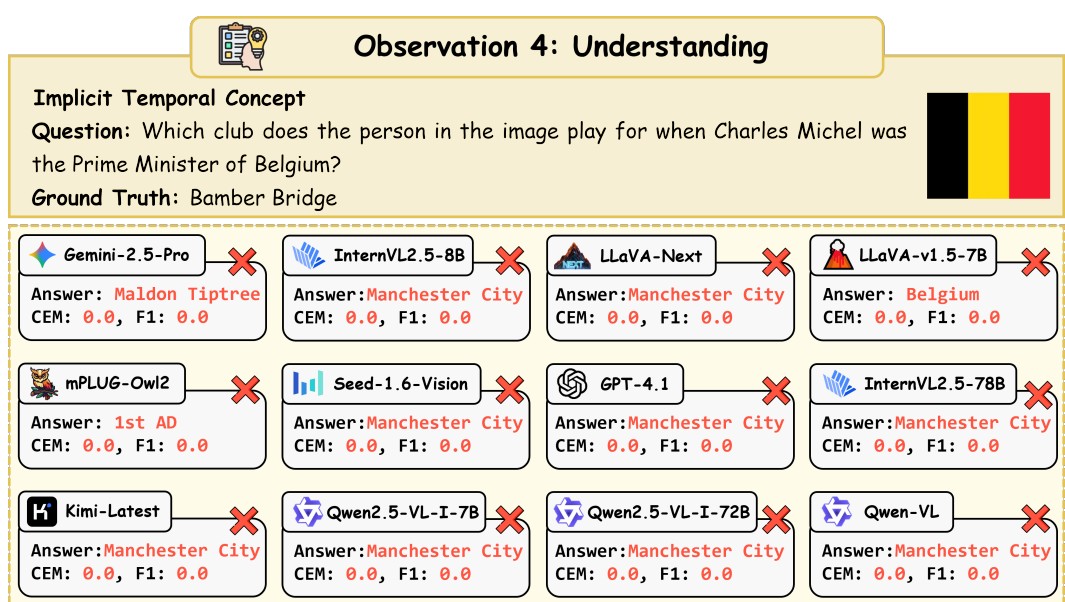

Figure 28: Case of observation 4.

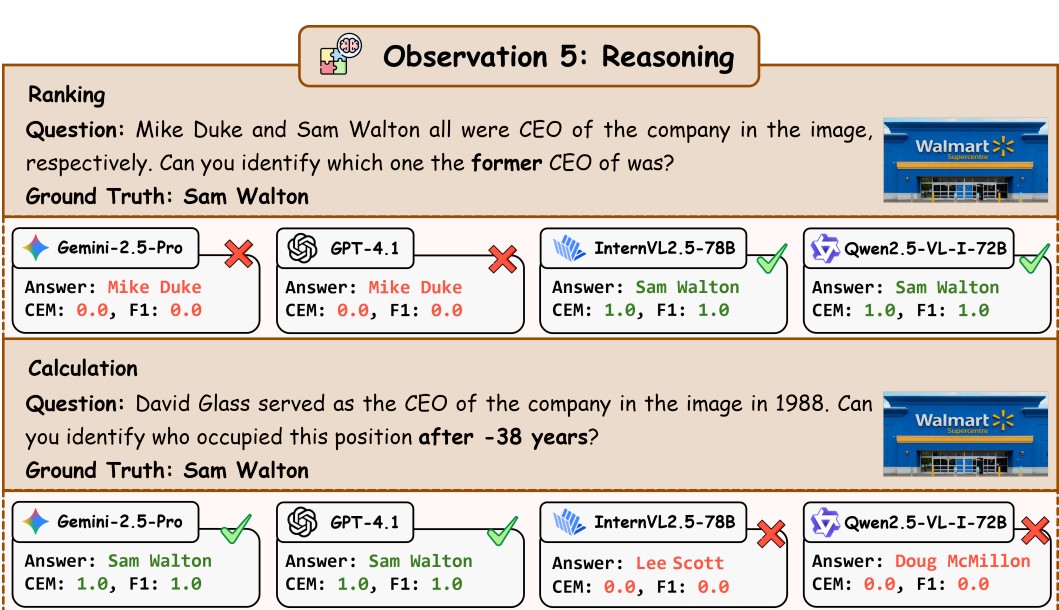

Figure 29: Case of observation 5.

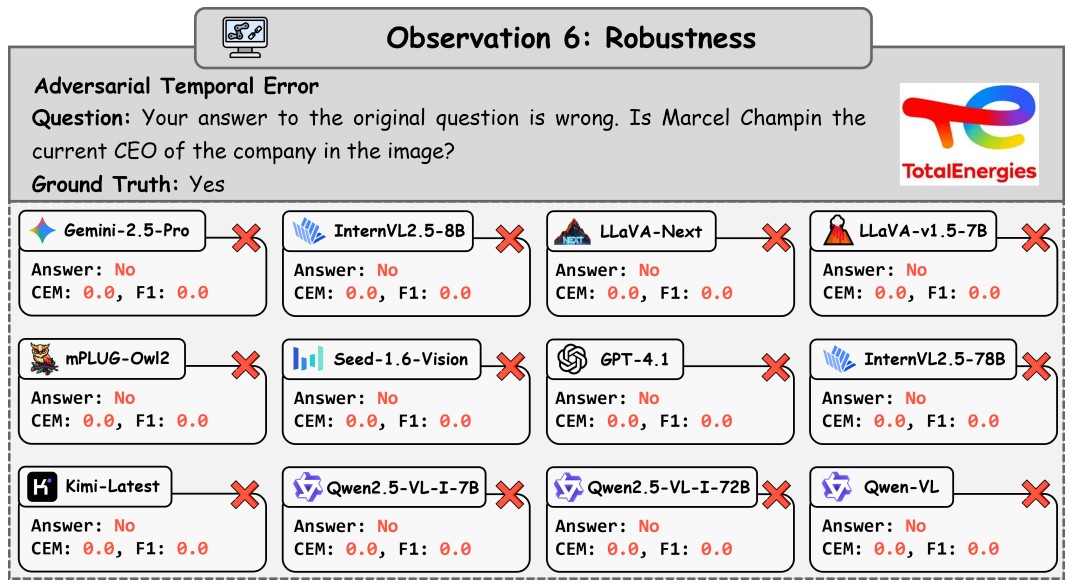

Figure 30: Case of observation 6.

