# OpenReview forum: "MINED: Probing and Updating  with Multimodal Time-Sensitive Knowledge for Large Multimodal Models"
_ICLR.cc/2026/Conference — ICLR 2026 Conference Withdrawn Submission_

### Official Review · Reviewer_3CwJ · 2025-10-23

**Soundness:** 3
**Presentation:** 3
**Contribution:** 3
**Rating:** 6
**Confidence:** 2

**Summary:**

This paper proposes MINED, a comprehensive benchmark that evaluates temporal awareness along 6 key dimensions and 11 challenging tasks: cognition, awareness, trustworthiness, understanding, reasoning, and robustness. MINED is constructed from Wikipedia by two professional annotators, containing 2,104 time-sensitive knowledge samples spanning six knowledge types.

**Strengths:**

1. The paper addresses a highly underexplored area—temporal awareness in Large Multimodal Models
2. The MINED benchmark is multi-dimensional, encompassing 6 core capabilities with comprehensive benchmark design
3. The analysis of the experimental conclusions is very meticulous

**Weaknesses:**

1. In Section 5, both the knowledge-editing methods and the backbone models are rather outdated. I suggest the authors experiment with more recent backbone models such as Qwen2.5-VL and adopt newer knowledge-editing techniques.
2. To make the experimental conclusions easier to grasp, the authors should provide more detailed case studies, using them to elaborate on the seven observations they draw.

**Questions:**

Please refer to Weaknesses

---

> ### Author Response · Authors · 2025-11-23
> **Response to reviewer 3CwJ (1/2)**
>
> Dear Reviewer 3CwJ:
>
> Thank you for your positive feedback and valuable suggestions. We sincerely appreciate the time and effort you have dedicated to reviewing our work. Below, we meticulously provide responses to each of your comments and outline the modifications based on your suggestions. All revisions are highlighted in blue.
>
> ---
> > **W1: In Section 5, both the knowledge-editing methods and the backbone models are rather outdated.**
>
>
> - Our primary aim in selecting backbone models is to choose those with **poor pre-editing performance**, thereby **magnifying the performance gap**. LLaVA-v1.5-7B and Qwen-VL are selected for experimentation because they are the two worst-performing models in Table 3 (Page 6, line 274-291) and are also **widely utilized in the knowledge editing field**.
> - We add three advanced knowledge editing methods: **RECIPE, LTE, and LiveEdit[1, 2, 3]**. The results, shown below, indicate that while these updated baselines perform worse than FT-LLM, FT-VIS, and IKE on single editing, they show better performance on lifelong editing. For instance, the average scores of LTE (57.17) and LiveEdit (59.06) far surpass FT-LLM's score (53.95).
>
>
> > **Table 1: Performance of knowledge editing methods with LLaVA-v1.5-7B on MINED.**
>
>
> | Method | Cog. |  |  | Tru. |  | Und. | Rea. |  | Rob. | Avg |
> |:---|:---:|:---:|:---:|:---:|:---:|:---:|:---:|:---:|:---:|:---:|
> |  | T.A | T.I.A | T.S.A | P.U.D | F.U.D | I.T.C | R.K | C.A | A.T.E | Avg |
> | **Single Editing** |  |  |  |  |  |  |  |  |  |  |
> | RECIPE | 46.21 | 44.00 | 60.92 | 43.96 | 42.25 | 39.22 | 77.56 | 64.67 | 47.70 | 51.83 |
> | LTE | 95.48 | 91.82 | 99.57 | 10.34 | 15.04 | 82.55 | 79.70 | 92.93 | 70.83 | 70.92 |
> | LiveEdit | 65.14 | 65.84 | 83.66 | 50.73 | 48.24 | 55.59 | 91.40 | 77.99 | 54.37 | 65.88 |
> | **Lifelong Editing** |  |  |  |  |  |  |  |  |  |  |
> | RECIPE | 46.00 | 44.14 | 60.46 | 44.04 | 42.38 | 39.38 | 75.61 | 49.64 | 46.17 | 49.76 |
> | LTE | 53.27 | 58.73 | 74.69 | 10.34 | 15.04 | 67.55 | 79.70 | 83.90 | 71.35 | 57.17 |
> | LiveEdit | 54.74 | 50.57 | 71.04 | 51.70 | 50.07 | 45.06 | 85.42 | 68.64 | 54.30 | 59.06 |
>
>
> [1] Lifelong knowledge editing for llms with retrieval-augmented continuous prompt learning.(EMNLP24)
>
> [2] Learning to edit: Aligning llms with knowledge editing.(ACL24)
>
> [3] Lifelong knowledge editing for vision language models with low-rank mixture-of-experts.(CVPR25)

---

> ### Author Response · Authors · 2025-11-23
> **Response to reviewer 3CwJ (2/2)**
>
> > **W2: To make the experimental conclusions easier to grasp, the authors should provide more detailed case studies, using them to elaborate on the seven observations they draw.**
>
> Thank you for your meaningful suggestion. We have provided the following clarifications for each detailed case analysis of observations:
>
> - **Obs 1: LMMs exhibit improved cognitive performance when queries are framed as timestamp-aware task.** We use the outputs of the same attribute on the temporal interval-aware and timestamp-aware tasks to illustrate this conclusion, finding that LMMs perform better when faced with the timestamp-aware task. `(Case in Appendix M, Page 36, Figure 25)`
>
> - **Obs 2: LMMs are vulnerable to temporal misaligned context, especially from past temporal misaligned contexts.** We use the outputs of the same knowledge when utilizing past misaligned context and future misaligned context to illustrate this conclusion, finding that LMMs are more easily affected by the past misaligned context. `(Case in Appendix M, Page 36, Figure 26)`
>
>
> - **Obs 3: LMMs are better at rejecting questions with unanswerable future dates than those with past dates.** We use the outputs of the same entity when faced with a past unanswerable date and a future unanswerable date to illustrate this conclusion, finding that LMMs perform better on the future unanswerable date task. `(Case in Appendix M, Page 37, Figure 27)`
>
> - **Obs 4: All LLMs perform terribly on tasks involving implicit temporal concepts.** We use examples where all 12 LMMs fail the implicit temporal concept task to illustrate this conclusion, finding that LMMs perform extremely poorly on this task. `(Case in Appendix M, Page 37, Figure 28)`
>
> - **Obs 5: Open-source LMMs demonstrate stronger performance on simpler ranking task, whereas closed-source LMMs excel in more complex calculation task.** We use the outputs of open-source and closed-source models when facing the same ranking and calculation tasks to illustrate this conclusion, finding that open-source models perform better on the ranking task, and closed-source models perform better on the calculation task. `(Case in Appendix M, Page 38, Figure 29)`
>
> - **Obs 6: Current LMMs demonstrate limited adversarial robustness against temporal errors.** We use examples where all 12 LMMs fail the adversarial temporal error task to illustrate this conclusion, finding that LMMs perform extremely poorly on this task. `(Case in Appendix M, Page 38, Figure 30)`
>
> - **Obs 7: More recent LMMs exhibit better temporal awareness performance.** This conclusion is primarily illustrated by comparing the average scores in Table 3 (Page 6, Line 274-291), where we find that the overall performance of closed-source models far surpasses that of open-source models. This finding holds true across all three evaluation paradigms (CEM `Page 16, Table 9, Line 835-861`, F1-score `Page 17, Table 10, Line 864-890`, and LLM as judge `Page 34, Table 12, Line 1818-1835`). Furthermore, we provide detailed examples for 11 tasks in Appendix E `(Page 18-21, Figure 9-19)`, enabling a detailed comparison of the specific outputs of the two model types.
>
>
> **We hope these cases help you and readers grasp the 7 observations we provide more easily**. Hope our clarification could resolve your concern.

---

> ### Author Response · Authors · 2025-11-27
> **Please tell us if any concern remains**
>
> Dear Reviewer 3CwJ,
> ﻿
>
>
> We truly appreciate the effort you have invested in reviewing this paper. We have submitted a **detailed rebuttal** addressing each of your points.
>
> **If you find that any explanation is still insufficient or could benefit from further clarification, please tell us and we would be happy to elaborate.**
>
>
> Thanks for your consideration!
>
>
> Authors of Paper 365

---

### Official Review · Reviewer_cs3Q · 2025-10-28

**Soundness:** 4
**Presentation:** 3
**Contribution:** 4
**Rating:** 8
**Confidence:** 3

**Summary:**

This paper introduces MINED, a new benchmark designed to systematically evaluate the performance of multimodal large language models on time-sensitive questions and dynamic knowledge updating capabilities. The benchmark assesses six dimensions. Using MINED, the paper evaluates several recent models and further investigate the performance of classical knowledge editing methods on updating time-sensitive knowledge.

**Strengths:**

MINED offers a comprehensive benchmark of dynamic knowledge understanding in LMMs, with extensive evaluations of recent models and a systematic study of knowledge-editing methods for time-sensitive updates. By jointly leveraging text and images to probe temporal awareness and misalignment, it takes a crucial step toward realistic multimodal evaluation.

**Weaknesses:**

1. Although the evaluation dimensions have increased, it seems that the proposed dataset is not more challenging than other datasets.
2.Missing inter-annotator agreement (e.g., Cohen’s κ) and conflict resolution details. These should be reported for transparency.
3.The automatically generated misaligned contexts might encode stylistic artifacts; human-written or multi-source variants are needed.

**Questions:**

1. Is the dataset updated over time?
2. The maximum answer length is 13, but the average answer length is only 2. Are most of the answers single words?

---

> ### Author Response · Authors · 2025-11-23
> **Response to reviewer cs3Q (1/2)**
>
> Dear Reviewer cs3Q:
>
> Thank you for your positive feedback and valuable suggestions. We sincerely appreciate the time and effort you have dedicated to reviewing our work. Below, we meticulously provide responses to each of your comments and outline the modifications based on your suggestions. All revisions are highlighted in blue.
>
> ---
> > **W1: Although the evaluation dimensions have increased, it seems that the proposed dataset is not more challenging than other datasets.**
>
>
> - **The evaluation of existing benchmarks suffers from serious deficiencies:**
>     - Benchmarks such as TimeQA, MenatQA, TempReason, DyKnow, and EvoWiki focus primarily on evaluating the Cognition dimension of time-sensitive knowledge. Their deficiencies in other dimensional evaluations result in inadequate difficulty and depth for assessing overall temporal awareness.
>     - Although EvolveBench involves more dimensions, it focuses solely on the text domain and lacks the visual localization process. Consequently, its difficulty is far lower than MINED's complex chained evaluation, which consists of **visual localization (locate visual entities through visual information) and knowledge recall**.
>     - MINED's evaluation employs **Prompt Agreement (Page 5-6, Line 268-272)**, requiring the model to give a correct response even when using the generalization question and image.
>
> - **The overall scores of the models on MINED are low:** We evaluate 15 mainstream LMMs, finding that the overall average CEM score across all tasks is **only 38.76%**. Even the best-performing model, Gemini-2.5-Pro, only achieves 63.07%, clearly indicating that LMMs are far from mastering the capabilities assessed by MINED.
>
> - **Fine-grained task scores:** Due to the text domain focus of most benchmarks, we provide a qualitative and rough comparison by citing the best model's performance across shared dimensions. The scores (e.g., **MINED's T.A.=37.58% vs. DyKnow=80%; MINED's I.T.C.=18.73% vs. EvolveBench=75%**) clearly demonstrate that MINED is more challenging than other datasets.
>
>
>
>  ---
> > **W2: Missing inter-annotator agreement (e.g., Cohen’s κ) and conflict resolution details. These should be reported for transparency.**
>
>
>
> - **(1) Measures for handling different opinions:** When the two annotators encounter a conflict, **we only retain the data that both agree upon**.
> - **(2) Inter Annotator Agreement numebers:** We calculate `Cohen's Kappa = 0.8273` using the formula $k = 1 - \frac{1 - p_o}{1 - p_e}$, where $p_o$ denotes the observed agreement ratio between experts and $p_e$ denotes the hypothetical probability of chance agreement. Samples meeting the aforementioned screening criteria are classified as positive samples; others are negative samples. The screening results are as follows:
>
> > **Table 1: Inter annotator agreement numebers.**
>
> |  | positive (annotator A) | negative (annotator A) | Total |
> |:---:|:---:|:---:|:---:|
> | positive (annotator B) | 455 | 19 | 474 |
> | negative (annotator B) | 18 | 120 | 138 |
> | Total | 473 | 139 | 612 |

---

> ### Author Response · Authors · 2025-11-23
> **Response to reviewer cs3Q (2/2)**
>
> > **W3: The automatically generated misaligned contexts might encode stylistic artifacts; human-written or multi-source variants are needed.**
>
>
> - **Synthetic data from GPT-4o:** In Dimension 2 Awareness, we use GPT-4o to generate the `Future and Past Misaligned Contexts`. Since this process may lead to semantic distortion and the introduction of bias, **we address these concerns before data synthesis**. Specifically, **we ensure semantic fidelity and avoid bias through mandatory task instructions and diverse task examples**, respectively. For example, one instruction is: `You must generate authentic and relevant descriptions based on the provided information`.
> - **Human studies for verifying data quality:** We randomly sample 20 data points from both the F.M.C. and P.M.C. tasks, requiring two annotators to manually write misaligned contexts for each. The annotators then compare the GPT-4o generated contexts against the human-written contexts to check for semantic distortion and bias, assigning a score between 0 and 10. A higher score indicates better quality for the GPT-4o context. **(Front-end examples used in the human study in Appendix I.2, Figure 23-24)**
>
> > **Table 2: Human studies for synthetic data and manual writing data.**
>
> | | F.M.C | P.M.C |
> | :--- | :---: | :---: |
> | **Annotator A** | | |
> | GPT-4o generated contexts vs Manual writing 1 | 9.75 | 9.70 |
> | GPT-4o generated contexts vs Manual writing 2 | 9.57 | 9.65 |
> | Mean-variance | 9.66±0.13 | 9.68±0.04 |
> | **Annotator B** | | |
> | GPT-4o generated contexts vs Manual writing 1 | 9.70 | 9.68 |
> | GPT-4o generated contexts vs Manual writing 2 | 9.82 | 9.70 |
> | Mean-variance | 9.76±0.08 | 9.69±0.01 |
>
>
> - **Experimental verification:** We test the performance change on these 20 data points using contexts generated by GPT-4o and manually written contexts from the two annotators as misaligned contexts, respectively. We find **the resulting performance fluctuation to be negligible**, confirming the high quality of the GPT-4o generated contexts.
>
>
> > **Table 3: Performance using synthetic data and manual writing data separately.**
>
>
> | | F.M.C | P.M.C |
> | :--- | :---: | :---: |
> | **LLaVA-v1.5-7B** | | |
> | GPT-4o generated contexts | 54.77 | 51.65 |
> | Manual writing 1 | 54.53 | 50.54 |
> | Manual writing 2 | 54.95 | 51.93 |
> | Mean-variance | 54.74±0.17 | 51.24±0.78 |
> | **GPT4.1** | | |
> | GPT-4o generated contexts | 66.52 | 91.91 |
> | Manual writing 1 | 66.86 | 91.24 |
> | Manual writing 2 | 66.35 | 91.16 |
> | Mean-variance | 66.61±0.24 | 91.20±0.47 |
>
>
> ---
> > **Q1: Is the dataset updated over time?**
>
>
>
> **Yes, MINED is updated over time**, with quarterly updates planned. We introduce MINED's evolvability in Appendix B.1 (Page 15, Line 772-792) and now reiterate the implementation process.
> - Leveraging existing MINED subject $S$ data, we retrieve corresponding Wikipedia text data offline (e.g., searching "Lionel Messi").
> - For club affiliation information, we extract information from Wikipedia's career sections using GPT-4o with strict parsing rules(the career field contains Lionel Messi's club affiliation information).
> - Newly extracted club data is compared against MINED's current records, triggering updates when discrepancies occur.
>
>
>
> ---
> > **Q2: The maximum answer length is 13, but the average answer length is only 2. Are most of the answers single words?**
>
> **Yes, most answers consist of one to two words**. Specific details are available in the case study. `(Appendix E, Page 18-21, Figure 9-19)`
> - Adversarial Temporal Error (Robustness) task's answer is **Yes/No**.
> - Past Unanswerable Date and Future Unanswerable Date (Trustworthiness) task's answer is **Unknown**.
> - The answers for other tasks mostly consist of **names**, such as person names, company names, and team names.
>
>
> **Hope our clarification could resolve your concern.**

---

> ### Author Response · Authors · 2025-11-27
> **Please tell us if any concern remains**
>
> Dear Reviewer cs3Q,
> ﻿
>
>
> We truly appreciate the effort you have invested in reviewing this paper. We have submitted a **detailed rebuttal** addressing each of your points.
>
> **If you find that any explanation is still insufficient or could benefit from further clarification, please tell us and we would be happy to elaborate.**
>
>
> Thanks for your consideration!
>
>
> Authors of Paper 365

---

### Official Review · Reviewer_oQfe · 2025-10-31

**Soundness:** 3
**Presentation:** 3
**Contribution:** 2
**Rating:** 4
**Confidence:** 3

**Summary:**

This paper introduces MINED, a comprehensive benchmark designed to evaluate large multimodal models’ (MLLMs) ability to understand and update time-sensitive knowledge.
Unlike prior static or text-only temporal benchmarks, MINED focuses on multimodal temporal awareness—how models perceive, reason about, and update factual knowledge that evolves over time.

The benchmark contains 2,104 time-sensitive knowledge samples and 4,208 questions, spanning six capability dimensions (Cognition, Awareness, Trustworthiness, Understanding, Reasoning, and Robustness) and six knowledge types (e.g., sport, organization, company).
Extensive evaluations over 15 popular LMMs reveal that even state-of-the-art closed-source models (e.g., Gemini-2.5-Pro) struggle with implicit and misaligned temporal knowledge.
Furthermore, the authors explore knowledge editing methods (FT-LLM, FT-VIS, MEND, SERAC, IKE) to update outdated time-sensitive facts in LMMs, demonstrating that single-editing is effective, while lifelong editing suffers from catastrophic forgetting.

Overall, MINED aims to bridge temporal reasoning, multimodal grounding, and model updating—offering a unified platform for evaluating time-sensitive understanding and editing in LMMs.

**Strengths:**

1. **Comprehensive benchmark design.**
   - The benchmark is organized into six orthogonal capability dimensions, forming a well-structured framework that captures not only factual recall but also temporal awareness, reasoning, and robustness.
   - The inclusion of *temporal misalignment* and *unanswerable-date* subtasks realistically simulates real-world temporal inconsistencies that often occur in dynamic factual knowledge.
2. **High-quality dataset and clear evaluation protocol.**
   - All data are manually verified, and the benchmark supports *evolvability* through quarterly Wikipedia updates, which ensures long-term relevance.
   - The proposed *Prompt Agreement* scheme is a thoughtful methodological design that effectively mitigates prompt-induced variance during evaluation.
3. **Strong experimental depth and breadth.**
   - Evaluation across 15 large multimodal models (both open- and closed-source) provides rich comparative insights into current model limitations.
   - The experimental analysis includes detailed error breakdowns, cross-model comparisons, and multiple editing paradigms, offering a well-rounded understanding of temporal sensitivity in LMMs.

**Weaknesses:**

1. **Limited novelty and motivation.**
   - The paper mainly focuses on dataset construction, while the necessity and motivation for studying this particular task could be discussed more clearly.
   - There is little exploration of potential methodological improvements or model-side strategies for enhancing temporal sensitivity beyond the dataset itself.
2. **Relatively small scale of data.**
   - The benchmark includes 4,208 questions divided into seven categories, with only 450 unique images.
   - It remains somewhat unclear whether such a dataset size is sufficient to comprehensively probe multimodal temporal sensitivity, especially given the complexity of real-world temporal reasoning.
3. **Brief analysis of lifelong editing.**
   - The section on “lifelong editing” is rather concise, and the underlying causes of catastrophic forgetting are not deeply analyzed.
   - A more systematic examination of how editing interacts with multimodal representations would strengthen this part.
4. **Evaluation metric limitations.**
   - Heavy reliance on the *Correct Exact Match (CEM)* metric may underestimate partial correctness.
   - Although F1 scores are reported in the appendix, the main results still depend primarily on strict matching, which might not fully reflect model understanding.

**Questions:**

1. **On the necessity of multimodal extension:**
    The motivation for extending temporal reasoning to multimodal settings could be further clarified.
    If the underlying language model in a multimodal system already possesses temporal sensitivity, does the multimodal extension inherently inherit such ability?
    What makes the multimodal setting particularly challenging or unique in this context?
2. **On the temporal validity of images:**
    Would it be more meaningful to incorporate visual changes over time—such as variations in a person’s appearance (childhood, adulthood, aging) or environmental transformations (seasons, locations)?
    What is the essential difference between the temporal sensitivity problem in MINED and that in purely text-based temporal knowledge benchmarks?
3. **On the simplicity of the knowledge representation:**
    The benchmark relies on a quadruple-based knowledge structure $$(S,H,P,A)$$.
    While this design enables systematic probing along the six dimensions introduced in Section 3.1, it might oversimplify the complexity of temporal evolution in real-world multimodal data.
    Could the authors discuss whether this abstraction limits the benchmark’s ecological validity, and how future work might move toward more realistic, context-rich temporal scenarios?

---

> ### Author Response · Authors · 2025-11-23
> **Response to reviewer oQfe (1/5)**
>
> Dear Reviewer oQfe:
>
> Thank you for your positive feedback and valuable suggestions. We sincerely appreciate the time and effort you have dedicated to reviewing our work. Below, we meticulously provide responses to each of your comments and outline the modifications based on your suggestions. All revisions are highlighted in blue.
>
> ---
> > **W1: Limited novelty and motivation. The paper mainly focuses on dataset construction, while the necessity and motivation for studying this particular task could be discussed more clearly. There is little exploration of potential methodological improvements or model-side strategies for enhancing temporal sensitivity beyond the dataset itself.**
>
>
> - **Reiteration of task's necessity and motivation:** We first emphasize the core motivation and necessity of our work, which is also recognized by other reviewers.
>     - **(1) Current evaluation's deficiencies:** Existing benchmarks, such as TimeQA and TempReason, primarily focus on temporal reasoning in the pure text domain; even multimodal ones like LiveVQA lack a systematic evaluation of time-sensitive factual knowledge and fail to cover real-world issues such as temporal misalignment, conflicting information, and outdated knowledge.
>     - **(2) Filling the evaluation gap for multimodal time-sensitive knowledge:** We propose MINED, a novel, comprehensive, multi-dimensional benchmark specifically designed to evaluate the time-sensitive knowledge capabilities of large-scale LMMs. We refine temporal awareness into 6 key dimensions and 11 challenging subtasks, which provides a multi-dimensional and fine-grained diagnosis of LMMs' understanding of time-sensitive knowledge.
>     - **(3) Affirmation from other reviewers:** Other reviewers consistently affirm our motivation. `Reviewer 3CwJ's comment:` "addresses a highly underexplored area"; `Reviewer cs3Q's comment:` "it takes a crucial step toward realistic multimodal evaluation".
>
> Our explicit and highly necessary motivation is to build a high-quality, comprehensive benchmark that addresses the lack of systematic evaluation for multimodal time-sensitive knowledge. Following your suggestion, we clarify this motivation further in `Related Works (Page 3, Line 136-140)` section, hoping to improve the clarity of our work's positioning.
>
>
> - **Clarification regarding methodological improvements and future directions:**
>     - Our core contribution is the construction of a high-quality, comprehensive benchmark for multimodal time-sensitive knowledge. We deeply analyze the fundamental weaknesses of existing LMMs in handling temporal knowledge, including their vulnerability to misaligned context `(Page 6, Obs 2)`, poor understanding of implicit temporal concepts `(Page 7, Obs 4)`, extremely low robustness to temporal errors `(Page 7, Obs 6)`, and issues of knowledge obsolescence, distortion, and hallucination `(Page 7-8, Exp 3)`.
>     - In addition to evaluating multimodal temporal knowledge and identifying LMM weaknesses, we **verify the effectiveness of knowledge editing for updating this knowledge** in a single editing scenario, **providing direction for future work**.
>     - By revealing LMM deficiencies, MINED clearly guides future research. Poor performance in Awareness suggests **focusing on distinguishing internal and external context consistency**. Low Understanding performance emphasizes the **need for better semantic comprehension of implicit temporal concepts**. Finally, poor Robustness prompts the development of **stronger self-correction and adversarial robustness mechanisms**.
>
> Thus, although our work focues on experimental evluations, our observations and explorations offer improved directions for subsequent method design, elevating our work from mere **"evaluation"** to **"guiding future research."** Per your suggestion, we modify the original `Conclusion` section to `Conclusion and Discussion (Page 9-10, Line 483-494)` in the manuscript to clarify this crucial guiding contribution.

---

> ### Author Response · Authors · 2025-11-23
> **Response to reviewer oQfe (2/5)**
>
> > **W2: Relatively small scale of data. The benchmark includes 4,208 questions divided into seven categories, with only 450 unique images. It remains somewhat unclear whether such a dataset size is sufficient to comprehensively probe multimodal temporal sensitivity, especially given the complexity of real-world temporal reasoning.**
>
> - **(1) Data scale falls within a reasonable range:** Existing benchmarks show a range of sizes: smaller ones like DynKnow(EMNLP24) and EvolveBench(ACL25) contain **130 and 1,640 samples**, while larger ones like EvoWiki(ACL25) and UnseenTimeQA(ACL25) contain 10,264 and 10,800 samples. Compared to these, our benchmark, containing 4,208 questions, **falls into an intermediate range and represents a reasonable data scale**.
>
>
> - **(2)** The evaluation of time-sensitive knowledge **focuses more on depth than on breadth**: MINED designs 6 evaluation dimensions and 11 subtasks across 6 knowledge domains (Country, Sport, Company, University, Organization, and Competition). **This comprehensive structure ensures a deep evaluation of every sample, allowing for a thorough investigation into the temporal awareness capabilities of LMMs**.
>
> - **(3) Introducing generalization images ensures the diversity of visual knowledge:** Despite having only 450 unique images, the use of both the **original image and the generalization image for each sample during evaluation** (Page 5-6, Line 268-272) ensures visual knowledge diversity and prevents bias in the assessment results.
>
>
> ---
> > **W3: Brief analysis of lifelong editing. The section on “lifelong editing” is rather concise, and the underlying causes of catastrophic forgetting are not deeply analyzed. A more systematic examination of how editing interacts with multimodal representations would strengthen this part.**
>
>
> - **(1) Core positioning of the work is to evaluate and to indicate directions for subsequent research:** We focus on evaluating multimodal time-sensitive knowledge and propose MINED, a multi-dimensional and comprehensive benchmark. Our work reveals the major deficiencies **(in W1)** of existing LMMs in temporal awarenes.
>
> - **(2) Knowledge editing serves as an exploratory tool:** The purpose of introducing knowledge editing is to explore and validate its feasibility in updating time sensitive knowledge of LMMs, **rather than to delve into the mechanisms of knowledge editing algorithms themselves or the fine interactions of multimodal representations**.
>
>
> - **(3) Speculation of catastrophic forgetting:** We offer thoughts on catastrophic forgetting in lifelong editing. Knowledge editing methods typically use **a pipeline mode where they perform a single iteration on each editing sample until convergence before moving to the next**. This process leads to **overfitting the current sample**, which **strongly interferes with the model weights and prevents consistency across multiple edits**, thereby causing catastrophic forgetting of previously edited samples and original knowledge.
>
>
> - Verification of sequential editing: We employ a sequential editing setup for a simple verification experiment. Specifically, when `gap=5`, we continuously edit samples 1 through 6, observe the performance of sample 1, and then restore the model weights to their original state. In this setup, **adjusting the gap value allows us to observe how the degree of forgetting of previously edited knowledge changes as the number of edited samples increases**. We experiment with FT-LLM and LiveEdit[1] across six tasks. We observe that **editing performance gradually declines as the gap increases**, indicating that converging individually on each editing sample interferes with the performance of previously edited knowledge. `gap = 0 represents single editing`
>
> > **Table 1: Performance of sequential editing with LLaVA-v1.5-7B.**
>
> | Gap | Cog. |  |  | Und. | Rea. |  |  |
> |:---|:---:|:---:|:---:|:---:|:---:|:---:|:---:|
> |  | T.A | T.I.A | T.S.A | I.T.C | R.K | C.A |
> | FT-LLM |  |  |  |  |  |  |
> | gap = 0 | 100.00 | 100.00 | 100.00 | 100.00 | 100.00 | 100.00 |
> | gap = 10 | 83.36 | 72.75 | 62.36 | 67.76 | 60.04 | 67.39 |
> | gap = 20 | 76.36 | 69.25 | 58.47 | 59.56 | 54.60 | 62.60 |
> | gap = 50 | 70.02 | 68.25 | 52.22 | 53.52 | 43.11 | 53.88 |
> | LiveEdit |  |  |  |  |  |  |
> | gap = 0 | 69.74 | 72.00 | 69.06 | 67.30 | 89.42 | 73.29 |
> | gap = 10 | 66.11 | 55.50 | 52.89 | 56.86 | 86.04 | 68.36 |
> | gap = 20 | 62.94 | 54.25 | 50.31 | 55.19 | 73.04 | 64.64 |
> | gap = 50 | 58.20 | 42.15 | 43.01 | 52.52 | 68.36 | 59.71 |
>
> [1] Lifelong Knowledge Editing for Vision Language Models with Low-Rank Mixture-of-Experts.(CVPR25)

---

> ### Author Response · Authors · 2025-11-23
> **Response to reviewer oQfe (3/5)**
>
> > **W4: Evaluation metric limitations. Heavy reliance on the Correct Exact Match (CEM) metric may underestimate partial correctness. Although F1 scores are reported in the appendix, the main results still depend primarily on strict matching, which might not fully reflect model understanding.**
>
>
>
> - To ensure comprehensive evaluation of **semantic understanding**, we adopt the **"LLM as a judge"** paradigm. We use **GPT-4o** to semantically score the open-source models' **output against the ground truth**, assigning a score between 0 and 1, where a higher score indicates closer semantic proximity. (The task instructions and quantitative examples are in Appendix J, Page 34.)
>
> - The table reveals that the "LLM as a judge" paradigm captures the **semantic similarity between predictions and the ground truth more finely**, offering a more comprehensive assessment than CEM and F1-score. Crucially, the comparative performance trend among all models **remains consistent with our previous observations and explorations**, which confirms the reliability of our conclusions.
>
> > **Table 2: Overall Performance Comparison of MINED based on LLM judge.**
>
>
>
> | Models |  Cog. |  |  | Awa. |  | Tru. |  | Und. | Rea. |  | Rob. |  | |
> |:---|:---:|:---:|:---:|:---:|:---:|:---:|:---:|:---:|:---:|:---:|:---:|:---:|:---:|
> |  |  T.A | T.I.A | T.S.A | F.M.C | P.M.C | P.U.D | F.U.D | I.T.C | R.K | C.A | A.T.E | Avg |
> | **Open-source LMMs** | | | | |  | | | | | | | |
> | LLaVA-v1.5(7B) | 10.46 | 13.01 | 20.93 | 16.91 | 16.92 | 53.99 | 50.01 | 2.89 | 24.44 | 7.798 | 0.39 | **19.80** |
> | Qwen-VL(7B) | 20.20 | 25.44 | 55.46 | 18.64 | 19.05 | 81.27 | 70.17 | 9.10 | 39.52 | 27.22 | 0.00 | 33.27 |
> | mPLUG-Owl2(7B) | 16.50 | 20.06 | 56.93 | 52.92 | 49.24 | 12.00 | 44.42 | 5.38 | 52.31 | 23.79 | 6.12 | 30.86 |
> | LLaVA-Next-M.(7B) | 18.55 | 21.74 | 52.03 | 44.50 | 40.70 | 96.75 | 90.23 | 7.00 | 46.17 | 29.59 | 0.00 | 40.66 |
> | LLaVA-OV(7B) | 19.08 | 19.80 | 36.79 | 40.67 | 40.65 | 39.92 | 76.62 | 8.26 | 57.16 | 19.89 | 2.21 | 32.82 |
> | mPlug-Owl3(8B) | 16.51 | 18.90 | 41.89 | 40.63 | 38.72 | 98.07 | 99.76 | 6.31 | 46.33 | 13.30 | 3.66 | 38.50 |
> | MiniCPM-V2.6(8B) | 28.41 | 29.36 | 62.90 | 47.49 | 41.82 | 81.52 | 97.83 | 9.16 | 60.40 | 34.14 | 14.45 | 46.13 |
> | Qwen2-VL-I.(7B) | 26.37 | 27.62 | 44.76 | 30.00 | 24.44 | 99.52 | 99.76 | 10.60 | 56.62 | 27.26 | 9.90 | 41.53 |
> | InternVL2.5(8B) | 24.57 | 26.48 | 55.14 | 54.32 | 49.50 | 98.31 | 99.88 | 9.58 | 65.78 | 31.16 | 0.00 | 46.79 |
> | Qwen2.5-VL-I.(7B) | 26.48 | 27.78 | 53.21 | 51.75 | 45.83 | 99.64 | 99.76 | 9.83 | 48.07 | 34.64 | 17.78 | 46.80 |
> | **Closed-source LMMs** | | | | |  | | | | | | | |
> | Kimi-Latest | 33.69 | 34.56 | 78.89 | 76.91 | 74.44 | 72.12 | 86.59 | 12.33 | 54.11 | 52.93 | 6.38 | 53.00 |
> | Doubao-1.5-Vision-Pro | 40.25 | 37.80 | 80.59 | 81.41 | 78.06 | 93.12 | 100.00 | 10.07 | 40.07 | 44.26 | 12.24 | 56.17 |
> | Gemini-2.5-Pro | 41.60 | 62.04 | 90.40 | 88.94 | 89.62 | 79.22 | 96.28 | 20.84 | 47.47 | 84.78 | 39.52 | **69.20** |
> | GPT-4.1 | 41.16 | 47.41 | 87.47 | 84.99 | 85.27 | 65.36 | 91.41 | 13.63 | 37.41 | 66.81 | 17.58 | 58.05 |
> | Seed-1.6-Vision | 42.61 | 51.36 | 86.59 | 83.89 | 86.93 | 74.15 | 96.62 | 13.37 | 42.22 | 68.88 | 32.47 | **61.74** |
>
>
>
> ---
> > **Q1 (Part 1/2): On the necessity of multimodal extension: The motivation for extending temporal reasoning to multimodal settings could be further clarified. If the underlying language model in a multimodal system already possesses temporal sensitivity, does the multimodal extension inherently inherit such ability? What makes the multimodal setting particularly challenging or unique in this context?**
>
>
>
> - **Why is it necessary to extend temporal reasoning to multimodal settings?** Existing benchmarks mainly focus on temporal perception in the pure text domain. Given the complexity and diversity of real-world multimodal time-sensitive knowledge (e.g., a person's image across different contexts can hinder recall), we propose MINED. This benchmark addresses the current evaluation gap and systematically assesses LMMs' capability to handle this multimodal temporal knowledge.
>
> - **Does the multimodal extension inherently inherit the temporal sensitivity of the underlying language model?** We argue that the inheritance capability is limited because the multimodal temporal perception scenario is inherently more complex. Although the underlying LLMs encode rich time-sensitive knowledge, this knowledge is not fully or stably integrated when the multimodal setting introduces visual information that conflicts with the model's internal representations. **For instance, while the model recognizes Messi in a uniform, it may struggle to link an image of him in casual wear to that internal knowledge**.

---

> ### Author Response · Authors · 2025-11-23
> **Response to reviewer oQfe (4/5)**
>
> > **Q1 (Part 2/2): On the necessity of multimodal extension: The motivation for extending temporal reasoning to multimodal settings could be further clarified. If the underlying language model in a multimodal system already possesses temporal sensitivity, does the multimodal extension inherently inherit such ability? What makes the multimodal setting particularly challenging or unique in this context?**
>
> - **Why is the multimodal setting more challenging?** **(1)** Multimodal time-sensitive knowledge evaluation requires a complex process: identifying and locating the visual entity via image information, and then linking it with the correct time-sensitive knowledge. This complex process of **visual localization** and **knowledge recall** is entirely absent in pure text evaluation. **(2)** The same visual entity can be represented by diverse images (from different times and locations); these generalization images (Page 5-6, Line 268-272) make it more difficult for LMMs to connect with internal knowledge.
>
>
> ---
> > **Q2: On the temporal validity of images: Would it be more meaningful to incorporate visual changes over time—such as variations in a person’s appearance (childhood, adulthood, aging) or environmental transformations (seasons, locations)? What is the essential difference between the temporal sensitivity problem in MINED and that in purely text-based temporal knowledge benchmarks?**
>
>
>
> - **Is it more meaningful to incorporate visual changes?**
>     - **Yes**, incorporating visual changes is significant as it leads to a more comprehensive and compelling evaluation. We enforce this through `Prompt Agreement`(Page 5-6, Line 268-272): each knowledge item is combined pair-wise from four elements `("Question", "Generalization Question", "Image", and "Generalization Image")` to create four distinct inputs. The inclusion of the **"Image" and "Generalization Image" specifically accounts for environmental transformations**.
>     - The results for the four input groups (Q is question, GQ is generalization question, I is image, GI is generalization image) are presented in the table below. A key observation is that when using the same question but varying the input image, **the model's responses show score fluctuations across all three evaluation paradigms** (CEM, F1-score, and LLM as judge). This clearly indicates the significance of incorporating visual changes. (More experimental results are in Appendix K, Page 35, Table 13.)
>
>
> > **Table 3: Overall Performance Comparison of MINED based on prompt agreement.**
>
> | Input | Cog. |  |  | Awa. |  | Tru. |  | Und. | Rea. |  | Rob. | Avg |
> |:---|:---:|:---:|:---:|:---:|:---:|:---:|:---:|:---:|:---:|:---:|:---:|:---:|
> |  | T.A | T.I.A | T.S.A | F.M.C | P.M.C | P.U.D | F.U.D | I.T.C | R.K | C.A | A.T.E | Avg |
> | LLaVA-v1.5-7B |  |  |  |  |  |    |  |  |  |  |  |  |
> | **CEM** |  |  |  |  |  |    |  |  |  |  |  |  |
> | Q+I | 8.87 | 11.18 | 23.55 | 3.08 | 2.82 | 53.62 | 50.72 | 3.16 | 17.50 | 7.69 | 0.00 | 16.56 |
> | Q+GI | 7.07 | 9.20 | 22.56 | 2.18 | 2.84 | 48.79 | 49.75 | 1.29 | 18.75 | 6.49 | 0.52 | 15.40 |
> | GQ+I | 7.28 | 9.94 | 12.23 | 12.76 | 10.00 | 57.00 | 49.75 | 1.64 | 12.50 | 6.41 | 0.52 | 16.37 |
> | GQ+GI | 6.91 | 6.47 | 11.39 | 11.44 | 10.05 | 56.52 | 49.75 | 0.81 | 12.34 | 5.06 | 0.52 | 15.57 |
> | **F1-score** |  |  |  |  |  |   |  |  |  |  |  |  |
> | Q+I | 9.99 | 14.03 | 22.64 | 6.00 | 5.94 | 53.62 | 50.72 | 3.01 | 17.77 | 7.69 | 0.00 | 17.40 |
> | Q+GI | 7.86 | 11.65 | 22.36 | 4.93 | 5.69 | 48.79 | 49.75 | 2.21 | 18.75 | 6.49 | 0.52 | 16.27 |
> | GQ+I | 8.39 | 11.73 | 12.72 | 15.36 | 13.03 | 57.00 | 49.75 | 1.78 | 12.77 | 7.26 | 0.52 | 17.30 |
> | GQ+GI | 7.92 | 8.31 | 11.95 | 15.12 | 13.61 | 56.52 | 49.75 | 1.54 | 12.62 | 5.06 | 0.52 | 16.63 |
> | **LLM as judge** |  |  |  |  |  |   |  |  |  |  |  |  |
> | Q+I | 11.17 | 15.20 | 25.18 | 12.86 | 15.13 | 53.62 | 50.77 | 3.72 | 20.12 | 10.00 | 0.00 | 19.80 |
> | Q+GI | 9.15 | 13.54 | 25.78 | 12.73 | 14.66 | 48.79 | 49.75 | 3.21 | 21.35 | 7.65 | 0.52 | 18.83 |
> | GQ+I | 10.90 | 13.72 | 17.06 | 11.39 | 18.45 | 57.00 | 49.75 | 2.39 | 28.51 | 8.27 | 0.52 | 20.72 |
> | GQ+GI | 10.43 | 9.44 | 15.65 | 10.48 | 19.41 | 56.52 | 49.75 | 2.13 | 27.77 | 5.80 | 0.52 | 19.81 |
>
>
> - **What is the essential difference between MINED and purely text-based temporal knowledge benchmarks?** The essential difference is that MINED's evaluation requires both **visual localization** and **knowledge recall**, whereas purely text-based temporal knowledge benchmarks only require the latter. Additionally, MINED utilizes generalized visual knowledge, enabling a more comprehensive assessment.

---

> ### Author Response · Authors · 2025-11-23
> **Response to reviewer oQfe (5/5)**
>
> > **Q3: On the simplicity of the knowledge representation: The benchmark relies on a quadruple-based knowledge structure(S,H,P,A). While this design enables systematic probing along the six dimensions introduced in Section 3.1, it might oversimplify the complexity of temporal evolution in real-world multimodal data. Could the authors discuss whether this abstraction limits the benchmark’s ecological validity, and how future work might move toward more realistic, context-rich temporal scenarios?**
>
>
> - **Is knowledge representation simplified?**
>     - **The carefully designed quadruple is sufficient to fully express multimodal time-sensitive knowledge:** The (S, H, P, A) knowledge structure is meticulously designed to probe every piece of multimodal time-sensitive knowledge. This simple, elegant structure flexibly transforms into the six dimensional tasks discussed in the paper and **holds the potential for transformation into other types of dimensional assessments**.
>     - **Widespread recognition within the field of knowledge editing**: Triplet-formatted factual knowledge is universally adopted as the source for editing and evaluation in the knowledge editing field, regardless of whether it targets LLMs or LMMs. **Since this knowledge structure is both effective and widely recognized, it does not limit the benchmark’s ecological validity**.
>
> - **Thoughts on Future Work:** We completely agree with the reviewer's point that future work moves towards more realistic, context-rich temporal scenarios with greater ecological validity.  We believe potential directions include: (1) **Integrating richer time-dependent context by extending knowledge representation to incorporate trigger events and causal relations**, forming complex structures that simulate real-world knowledge evolution. (2) Exploring **Multi-hop Temporal Reasoning**, since current benchmarks focus on single-step retrieval, future work introduces tasks requiring multi-step reasoning chains. We add this content to Appendix L (Page 35, line 1864-1873) of the manuscript for reader reference.
>
>
> **Hope our clarification could resolve your concern.**

---

> ### Author Response · Authors · 2025-11-27
> **Please tell us if any concern remains**
>
> Dear Reviewer oQfe,
> ﻿
>
>
> We truly appreciate the effort you have invested in reviewing this paper. We have submitted a **detailed rebuttal** addressing each of your points.
>
> **If you find that any explanation is still insufficient or could benefit from further clarification, please tell us and we would be happy to elaborate.**
>
>
> Thanks for your consideration!
>
>
> Authors of Paper 365

---

### Official Review · Reviewer_bwJS · 2025-11-01

**Soundness:** 2
**Presentation:** 3
**Contribution:** 2
**Rating:** 4
**Confidence:** 4

**Summary:**

In this paper, the authors propose MINED, a comprehensive benchmark that evaluates temporal awareness along 6 key dimensions and 11
challenging tasks: cognition, awareness, trustworthiness, understanding, reasoning, and robustness. MINED is constructed from Wikipedia by two professional annotators, containing 2,104 time-sensitive knowledge samples spanning six knowledge types. The authors also evaluate more than 10 widely used LMMs in the proposed dataset.

**Strengths:**

1. In this paper, the authors propose MINED, a comprehensive benchmark that evaluates temporal awareness along 6 key dimensions and 11 challenging tasks: cognition, awareness, trustworthiness, understanding, reasoning, and robustness.

2. MINED is annotated by professional annotators, containing 2,104 time-sensitive knowledge samples spanning six knowledge types

**Weaknesses:**

1. In this paper, I didn't find how to build such a dataset.  The authors only mention that To construct the foundational data for MINED, we employ two professional annotators to gather time-sensitive knowledge from Wikipedia across six domains: Country, Sport, Company, University, Organization, and Competition.

2. Only two professional annotators are invovled in the annotation, How to deal with situations where two people have different opinions?

3. In this paper, how to control the quality of the proposed dataset is unknown

**Questions:**

NA

---

> ### Author Response · Authors · 2025-11-23
> **Response to reviewer bwJS (1/3)**
>
> Dear Reviewer bwJS:
>
> Thank you for your positive feedback and valuable suggestions. We sincerely appreciate the time and effort you have dedicated to reviewing our work. Below, we meticulously provide responses to each of your comments and outline the modifications based on your suggestions. All revisions are highlighted in blue.
>
> ---
> > **W1 (Part 1/2): In this paper, I didn't find how to build such a dataset.**
>
>
> We appreciate your feedback; providing further details on the data construction pipeline helps us refine our work and enables readers to better understand and reproduce our pipeline. We will clarify the **original data(Page 3, Line 157-166)** and **task data(Page 4-5, Line 168-227)** separately from the following aspects:
>
> - **(1)** Details of the **original data construction pipeline**:
>     - **Step 1**: We define Country, Sport, Company, University, Organization, and Competition as the target domains and subsequently **prompt GPT-4o to generate lists of suitable entity candidates for each**. The total number of entity candidates is 612.
>     - **Step 2**: Two annotators **manually search for information on every entity candidate via Wikipedia**. Data are retained only if they meet two criteria: **the entity must be visual** and accurately representable by an image (e.g., Lionel Messi), and **it must be time-sensitive**, meaning its attributes update over time (e.g., which team Lionel Messi currently plays for). Annotator A retains 473 entities, and annotator B retains 474 samples.
>     - **Step 3**: After discarding data where the two annotators disagree, we **manually collect** the following from Wikipedia for each remaining entry: the subject (S) (e.g., a person or visual entity name like Lionel Messi), the hypernym (H) (e.g., Lionel Messi’s hypernym is 'footballer'), the property (P) (e.g., the property between Lionel Messi and club is “play for”), a list of attribute values (A = [a1, a2, · · · , an], like a1="Paris Saint Germain F.C. | S:+2021-08-00 | E:+2023-06-30") for that property which change over time, and the original image (the entity image provided by Wikipedia). Each entity ultimately possesses **a quadruple (S, H, P, A) and an original image**.
>     - **Step 4**: To evaluate the temporal awareness ability of LMMs, a prerequisite is that the models possess perceptual capability, **meaning they must identify the evaluated entity from the image information**. We address this by constructing **manually written perception task question templates**, such as `What is the entity in this image? Answer with name.`, and randomly assign them to each entity data point, thereby creating a perception capability QA pair <perception task question, subject> for every piece of data.
>     - **Step 5**: We test the perception QA for each data point using 15 LMMs (e.g., LLaVA-v1.5-7B, Qwen-VL, and GPT-4.1). We consider LMMs to lack adequate perception ability for an entity if 10 of these models fail to identify the entity in the image. **To avoid interference with the subsequent temporal perception evaluation, we directly discard these failed entities**, ultimately retaining 255 entity samples.
>     - **Step 6**: We use the subject plus hypernym as search keywords to download entity images from Google. We then use CLIP to extract features from both the downloaded and original images and calculate their cosine similarity. After excluding samples with a similarity score of 1, **we select the top-1 resulting image as the generalization image**. (The function of generalization image is detailed in Section 4.1, Page 5-6, Line 268-272)
>     - **Step 7**: Each final data point comprises a quadruple (S, H, P, A), an original image, and a generalization image.
>
> **The detailed content and data construction flowchart are in Appendix H, Page 30, Line 1569-1612.**

---

> ### Author Response · Authors · 2025-11-23
> **Response to reviewer bwJS (2/3)**
>
> > **W1 (Part 2/2): In this paper, I didn't find how to build such a dataset.**
>
> - **(2)** Details of the **task data construction pipeline**: The task data primarily consists of **manually written task question templates**. We detail the construction process for each task data below. (Qualitative examples and chat templates in Appendix G, Page 24-29)
>     - **Dimension 1 Cognition:**
>         - **Time-Agnostic (T.A):** We first write **task question templates** for the 6 knowledge domains (Country, Sport, Company, University, Organization, and Competition), where the Sport templates, for instance, include `Which club does the {hypernym} in the image currently {property}?` and `The {hypernym} in the image currently {property}.` Subsequently, we fill the hypernym and property from the original data into the corresponding templates.
>         - **Temporal Interval-Aware (T.I.A):** We similarly write task question templates for each knowledge domain; for example, the Country templates are **Who was {property} the {hypernym} in the image from $T_{\text{start}}$ to $T_{\text{end}}$?** and **From $T_{\text{start}}$ to $T_{\text{end}}$, {property} the {hypernym} in the image was**.
>         - **Timestamp-Aware (T.S.A):** We write task question templates, such as the Company templates: **Who was {property} the {hypernym} in the image in $T_{\text{stamp}}$?** and **In $T_{\text{stamp}}$, {property} the {hypernym} in the image was**. Here, $T_{\text{stamp}}$ is a timestamp randomly selected from $T_{\text{start}}$ to $T_{\text{end}}$.
>     - **Dimension 2 Awareness:**
>         - **Future Misaligned Context (F.M.C):**: The construction of the question and answer aligns with the Timestamp-Aware task, utilizing the past timestamp $T_{\text{past}}$. Besides, we input (S, P, $a_{\text{current}}$) to prompt GPT-4o, which generates a relevant text description that serves as the `Future Misaligned Context`. The final task data `(Future Misaligned Context, Question, and Answer)` is processed as a single input unit.
>         - **Past Misaligned Context (P.M.C):**: Similarly to the Future Misaligned Context, we construct the QA using the current timestamp $T_{\text{current}}$ and generate the `Past Misaligned Context` using (S, P, $a_{\text{past}}$).
>     - **Dimension 3 Trustworthiness:**
>         - **Past Unanswerable Date (P.U.D):** Similarly to the Timestamp-Aware task, we randomly generate a `Past Unanswerable Date` for the attribute, which serves as $T_{\text{Past Unanswerable Date}}$.
>         - **Future Unanswerable Date (F.U.D):** Similarly to the Timestamp-Aware task, we randomly generate a `Future Unanswerable Date` for the attribute, which serves as $T_{\text{Future Unanswerable Date}}$.
>     - **Dimension 4 Understanding:**
>         - **Implicit Temporal Concept (I.T.C):** We use historical events to replace explicit time periods, such as the phrase `when Jeff Bezos served as CEO of Amazon`, which corresponds to the period `from July 5, 1994, to July 5, 2021` (Page 2, Figure 2). These historical events, which replace explicit time periods, are uniquely matched from the original data's attribute. **For instance, the time period when Jeff Bezos serves as CEO of Amazon, during which Lionel Messi plays exclusively for FC Barcelona, demonstrates temporal uniqueness.**
>     - **Dimension 5 Reasoning**:
>         - **Ranking (R.K):** We randomly select $a1$ and $a2$ from the original data's attribute list and write task question templates. For example, one template is: `{attribute-1} and {attribute-2} all were {property} the {hypernym} in the image, respectively. Can you identify which one the former {property} was?`.
>         - **Calculation (C.A):** We first randomly select $a1$ and $a2$ from the original data's attribute list. We then select two timestamps, $t_{\text{1}}$ and $t_{\text{2}}$, from $a1$'s and $a2$'s $T_{\text{start}}$ to $T_{\text{end}}$ ranges, respectively, and calculate the time difference $T_{\triangle}$. Finally, we write task question templates, such as: **{attribute} served as {property} the {hypernym} in the image in $t_{\text{1}}$. Can you identify who occupied this position after $T_{\triangle}$ years?**.
>     - **Dimension 6 Robustness**:
>         - **Adversarial Temporal Error (A.T.E):** We extract the QA pairs where **all models fail** the Cognition task. We then construct task question templates, such as: **Your answer to the original question is wrong. Was {attribute} {property} the {hypernym} in the image from $T_{\text{start}}$ to $T_{\text{end}}$?**, which require the model to output either Yes or No.
>
>
> We add all of this content in the manuscript to facilitate readers' understanding and reproduction of our process. (Appendix H, Page 30-31, Line 1613-1667)

---

> ### Author Response · Authors · 2025-11-23
> **Response to reviewer bwJS (3/3)**
>
> > **W2: Only two professional annotators are invovled in the annotation, How to deal with situations where two people have different opinions?**
>
>
> - **(1) Measures for handling different opinions:** As stated in step 2 of the original data construction pipeline (W1), when the two annotators encounter a conflict, **we only retain the data that both agree upon**.
> - **(2) Inter annotator agreement numebers:** We calculate `Cohen's Kappa = 0.8273` using the formula $k = 1 - \frac{1 - p_o}{1 - p_e}$, where $p_o$ denotes the observed agreement ratio between experts and $p_e$ denotes the hypothetical probability of chance agreement. Samples meeting the aforementioned screening criteria are classified as positive samples; others are negative samples. The screening results are as follows:
>
> > **Table 1: Inter annotator agreement numebers.**
>
> |  | positive (annotator A) | negative (annotator A) | Total|
> |:---:|:---:|:---:|:---:|
> | positive (annotator B) | 455 | 19 | 474 |
> | negative (annotator B) | 18 | 120 | 138 |
> | Total | 473 | 139 | 612 |
>
> ---
> > **W3: In this paper, how to control the quality of the proposed dataset is unknown**
>
>
> - **(1) Original data's quality:**
>     - **Textual data:** The text data for the entities are primarily **extracted manually from Wikipedia**, which is a **recognized source of high-quality data used by numerous benchmarks and datasets**, thereby ensuring the necessary data quality.
>     - **Visual data:** **Original image comes from Wikipedia** and is of extremely high quality. Generalization image, though downloaded from Google, also maintains extremely high quality because **it undergoes validation via cosine similarity against the original image's features**.
>     - **Cohen's Kappa:** As reported in W2, `Cohen's Kappa = 0.8273`, indicating almost perfect agreement. These results demonstrate that our manual collection process is highly consistent and reliable.
>     - **Human studies for verifying data quality:** We randomly select 100 data points for a human study involving two annotators. The task requires the annotators to check whether the content of the original data is authentic and reliable, and to assign a score between 0 and 10 to each data point; a higher score indicates higher data quality. **(Front-end examples used in the human study in Appendix I.1, Page 32, Figure 22)**
>
>
> > **Table 2: Human studies for original data.**
>
> |  | Annotator A | Annotator B | Mean-variance |
> |:---|:---:|:---:|:---:|
> | Score↑ | 9.74 | 9.78 | 9.76±0.03 |
>
>
> - **(2) Task data's quality:**
>     - **Task question templates:** All task question templates are **manually written and verified**, which ensures their extremely high quality.
>     - **Synthetic data from GPT-4o:** In Dimension 2 Awareness, we use GPT-4o to generate the `Future and Past Misaligned Contexts`. Since this process may lead to semantic distortion and the introduction of bias, **we address these concerns before data synthesis**. Specifically, we ensure semantic fidelity and avoid bias through mandatory task instructions and diverse task examples, respectively. For example, one instruction is: `You must generate authentic and relevant descriptions based on the provided information`.
>     - **Human studies for verifying data quality:** We randomly sample 20 data points from both the F.M.C. and P.M.C. tasks, requiring two annotators to manually write misaligned contexts for each. The annotators then compare the GPT-4o generated contexts against the human-written contexts to check for semantic distortion and bias, assigning a score between 0 and 10. A higher score indicates better quality for the GPT-4o context. **(Front-end examples used in the human study in Appendix I.2, Page 32-33, Figure 23-24)**
>
>
> > **Table 3: Human studies for task data.**
>
> | | F.M.C | P.M.C |
> | :--- | :---: | :---: |
> | **Annotator A** | | |
> | GPT-4o generated contexts vs Manual writing 1 | 9.75 | 9.70 |
> | GPT-4o generated contexts vs Manual writing 2 | 9.57 | 9.65 |
> | Mean-variance | 9.66±0.13 | 9.68±0.04 |
> | **Annotator B** | | |
> | GPT-4o generated contexts vs Manual writing 1 | 9.70 | 9.68 |
> | GPT-4o generated contexts vs Manual writing 2 | 9.82 | 9.70 |
> | Mean-variance | 9.76±0.08 | 9.69±0.01 |
>
>
> **Hope our clarification could resolve your concern.**

---

> ### Author Response · Authors · 2025-11-27
> **Please tell us if any concern remains**
>
> Dear Reviewer bwJS,
> ﻿
>
>
> We truly appreciate the effort you have invested in reviewing this paper. We have submitted a **detailed rebuttal** addressing each of your points.
>
> **If you find that any explanation is still insufficient or could benefit from further clarification, please tell us and we would be happy to elaborate.**
>
>
> Thanks for your consideration!
>
>
> Authors of Paper 365

---

### Author Response · Authors · 2025-11-23
**General Response**

Dear Reviewers,

**We sincerely appreciate your time, efforts, and insightful feedback on our work! We are delighted that all reviewers recognized the motivation, novelty, presentation, and experimental effectiveness of our study.**



We thank reviewers $\text{bwJS(R1)}$, $\text{oQfe(R2)}$, $\text{cs3Q(R3)}$, and $\text{3CwJ(R4)}$ for their insightful feedback. **Our work is recognized for addressing the highly underexplored area of temporal awareness in LMMs ($\text{R4}$), taking a crucial step toward realistic multimodal evaluation ($\text{R3}$)**. Reviewers uniformly commend the systematic and comprehensive nature of our benchmark MINED ($\text{R1}$, $\text{R2}$, $\text{R3}$, $\text{R4}$), highlighting its multi-dimensional design across six key capabilities ($\text{R1}$, $\text{R2}$, $\text{R4}$) and the high quality of the manually verified dataset ($\text{R1}$, $\text{R2}$). The inclusion of realistic temporal inconsistencies and quarterly updates is also highly praised for ensuring valid and evolving evaluation ($\text{R2}$, $\text{R3}$). Furthermore, reviewers acclaim the strong experimental depth ($\text{R2}$), specifically the extensive evaluation of 15 LMMs ($\text{R2}$, $\text{R3}$), the meticulous analysis of results ($\text{R2}$, $\text{R4}$), and the systematic exploration of knowledge-editing methods ($\text{R2}$, $\text{R3}$).


Below, we provide point-by-point responses to your comments and outline the revisions made to the manuscript based on your suggestions. All revisions are highlighted in blue. We have summarized our response as follows:

- **Clarification:**
    - Details of data construction pipeline, data annotation, and data quality. ($\text{R1,R3}$)
    - The motivation and necessity of MINED. ($\text{R2}$)
    - The effectiveness of visual changes. ($\text{R2}$)

- **New experiments and metric:**
    - Three new baselines for knowledge editing methods: RECIPE(EMNLP24), LTE(ACL24), and LiveEdit(CVPR25). ($\text{R4}$)
    - Analysis of catastrophic forgetting in lifelong editing based on sequential editing. ($\text{R2}$)
    - Introducing LLM as a judge paradigm for evaluation metrics. ($\text{R2}$)

- **New human studies:**
    - Quality of original data. ($\text{R1,R3}$)
    - Quality of synthesized data. ($\text{R3}$)



We warmly encourage you to review the results in the revised manuscript. Hope our response and additional experiments could address your concerns!

Furthermore, please allow us to reiterate the key contribution of our work: **MINED introduces a comprehensive benchmark designed to evaluate temporal awareness in Large Multimodal Models and investigate knowledge editing for updating time-sensitive knowledge**. We believe this contribution is crucial for advancing the field of multimodal time-sensitive knowledge, and we are truly grateful for your recognition of its significance.

Once again, we deeply appreciate the time and expertise you have shared with us. Your encouraging feedback motivates us to continue advancing this work for the broader community, and we are more than happy to add clarifications to address any additional recommendations and reviews from you！

Best regards,

Authors of Paper 365

---

### Author Response · Authors · 2025-11-30
**Summary for AC Consideration**

Dear Area Chair,

We sincerely appreciate your time and effort in handling our submission.

Unfortunately, we receive no reply from any reviewer during the discussion period. However, we believe the **positive scores** provided by **Reviewers cs3Q and 3CwJ** sufficiently indicate their approval of our work.

Although **Reviewers bwJS and oQfe** give negative scores, **we believe our rebuttal resolves their concerns**. Below, we provide a concise response to the primary worries raised by **Reviewers bwJS and oQfe**:




**(1) Benchmark construction pipeline and data quality** **(Reviewer bwJS)**:
- We provide a **detailed explanation and flowchart** for the construction pipeline in Appendix H `(Page 30-31, Line 1613-1667)`.
- To ensure data quality, we implement a series of **measures both before benchmark construction** and during verification via a post-construction **human study**.

**(2) Motivation and necessity of MINED** **(Reviewer oQfe)**:
- **Current evaluation's deficiencies:** Existing benchmarks prioritize text-based temporal reasoning (TimeQA, TempReason). Multimodal evaluations (LiveVQA) fail to systematically assess time-sensitive factual knowledge, omitting critical real-world challenges such as temporal misalignment and conflicting or outdated information.
- **Multimodal time-sensitive knowledge is more challenging:** Multimodal evaluation necessitates first locating the visual entity via image information, and then linking it with the correct time-sensitive knowledge. This complex chain of **visual localization** and **knowledge recall** is entirely absent in pure text assessment.





The corresponding clarifications and new experiments are summarized in our **General Response comment below** for your convenience.





Thank you again for your consideration.

Best regards,

Authors of Paper 365

---

### Note · Authors · 2025-12-16

I have read and agree with the venue's withdrawal policy on behalf of myself and my co-authors.